# Do LVLMs Truly Understand Video Anomalies? Revealing Hallucination via Co-Occurrence Patterns

**Menghao Zhang**[1][*]    **Huazheng Wang**[1][*]    **Pengfei Ren**[1]    **Kangheng Lin**[1]    **Qi Qi**[1]
**Haifeng Sun**[1][†]    **Zirui Zhuang**[1]    **Lei Zhang**[2]    **Jianxin Liao**[1]    **Jingyu Wang**[1][†]

[1]State Key Laboratory of Networking and Switching Technology,
Beijing University of Posts and Telecommunications, Beijing, China
[2]China Unicom, Beijing, China

{zhangmenghao, wanghz, rpf, linkangheng, qiqi8266, hfsun}@bupt.edu.cn
{zhuangzirui, liaojx, wangjingyu}@bupt.edu.cn; zhangl83@chinaunicom.cn

## Abstract

Large Vision-Language Models (LVLMs) pretrained on large-scale multimodal data have shown promising capabilities in Video Anomaly Detection (VAD). However, their ability to reason about abnormal events based on scene semantics remains underexplored. In this paper, we investigate LVLMs' behavior in VAD from a visual-textual co-occurrence perspective, focusing on whether their decisions are driven by statistical shortcuts between visual instances and textual phrases. By analyzing visual-textual co-occurrence in pretraining data and conducting experiments under different data settings, we reveal a hallucination phenomenon: *LVLMs tend to rely on co-occurrence patterns between visual instances and textual phrases associated with either normality or abnormality*, leading to incorrect predictions when these high-frequency objects appear in semantically mismatched contexts. To address this issue, we propose VAD-DPO, a direct preference optimization method supervised with counter-example pairs. By constructing visually similar but semantically contrasting video clips, VAD-DPO encourages the model to align its predictions with the semantics of scene rather than relying on co-occurrence patterns. Extensive experiments on six benchmark datasets demonstrate the effectiveness of VAD-DPO in enhancing both anomaly detection and reasoning performance, particularly in scene-dependent scenarios.

## 1 Introduction

Video Anomaly Detection (VAD) aims to automatically identify anomalous events in a given video, and has been widely applied in surveillance systems and smart city applications. Anomalies are typically defined as events or patterns that deviate from expected behaviors within a scene, meaning that the same visual action may appear normal in one context but anomalous in another. However, due to the limited availability of labeled anomaly data, most existing methods [35, 40, 46, 4, 41] tend to overfit the normal patterns present in the training data, and thus fail to interpret scene context and reason about anomalies.

With the rise of Large Vision-Language Models (LVLMs) [17, 18, 1], several studies explore the application of LVLMs to VAD, aiming to enhance their ability to reason about anomalies across diverse scenes. Early attempts [37, 36, 42, 39] leverage the general visual-language alignment capabilities of LVLMs to perform frame-level anomaly detection in a training-free manner. More

---

[*]Equal contribution.
[†]Corresponding author.

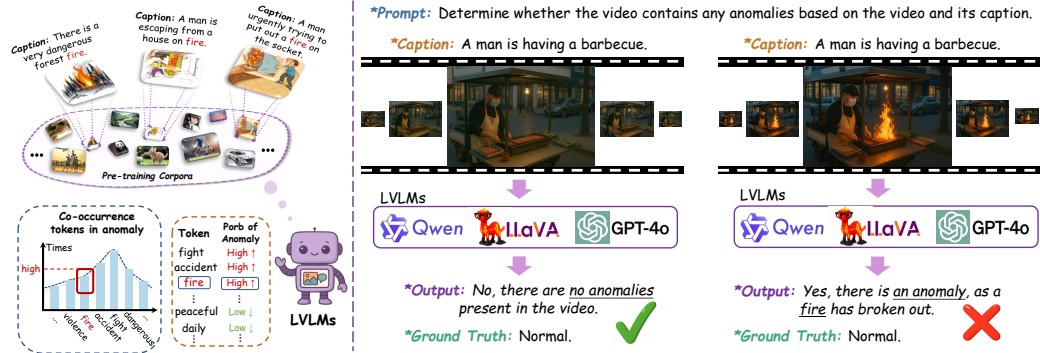

Figure 1: Illustration of hallucination in LVLM-based VAD. Although the input image and caption both describe a normal scenario ("a man is having a barbecue"), the LVLM incorrectly predicts an anomaly due to the presence of the word "fire." This behavior is driven by co-occurrence bias learned during pretraining, as visualized in the frequency statistics showing that "fire" is strongly associated with anomaly-related tokens in the pretraining corpus.

recent works [6, 29, 44] reformulate VAD as a video question answering task, using model-generated event descriptions to enhance interpretability. Despite the impressive performance of these LVLM-based VAD approaches, the underlying reasoning mechanisms behind their detection capabilities remain unclear. This gap motivates our study, as understanding how LVLMs reason about anomalies is critical for building safe and reliable VAD systems in real-world applications.

In this paper, we ask a fundamental question: ***Do LVLMs truly understand video anomalies?*** This question drives our investigation into whether LVLMs detect anomalies based on scene-aware understanding, or simply rely on superficial statistical shortcuts learned during pretraining. An ideal VAD model should interpret scene context and ground its predictions in visual evidence.

To investigate this question, we conduct an in-depth analysis of LVLMs' behavior in VAD from the perspective of visual-textual co-occurrence. Our study reveals a hallucination phenomenon: ***LVLMs tend to rely on co-occurrence patterns between visual instances and textual phrases associated with either normality or abnormality***. This reliance leads to incorrect predictions when high-frequency visual instances appear in semantically normal contexts. As shown in Figure 1, the model incorrectly predicts an anomaly in a scene where a man is having a barbecue, even though both the image and its caption describe a normal situation. To systematically examine this issue, we analyze co-occurrence patterns across multiple semantic levels, including objects, visual combinations, interactions, and temporal dynamics. Through both data analysis and controlled experiments, we show that such hallucinations are not isolated errors, but rather systematic failures rooted in the model's reliance on superficial statistical shortcuts instead of scene-aware reasoning about anomalies.

To address this issue, we propose **VAD-DPO**, a direct preference optimization framework tailored for VAD. The core idea is to formulate VAD as a pairwise preference optimization problem: by training on visually similar video clips with contrasting anomaly labels, the model is encouraged to distinguish semantic abnormality rather than rely on superficial statistical correlations. Specifically, we construct contrastive preference pairs in which the same object or scene context appears across clips, but only one constitutes an anomaly due to contextual differences. This preference-aligned fine-tuning process encourages the model to perform scene-aware reasoning, thereby mitigating hallucinations caused by spurious co-occurrence patterns.

In summary, our contributions are threefold:

• An underexplored problem in LVLM-based VAD is investigated, wherein a hallucination phenomenon is revealed from the perspective of visual-textual co-occurrence patterns. This suggests that anomaly detection is driven by statistical shortcuts rather than contextual reasoning.
• A preference-based optimization framework, VAD-DPO, is proposed to guide the model toward scene-aware reasoning and visually grounded anomaly detection by leveraging counter-example preference pairs with visually similar content but contrasting semantics.
• Extensive experiments conducted on six benchmark datasets demonstrate the superiority of VAD-

DPO in both anomaly detection and contextual reasoning. The improvements are particularly pronounced on the benchmarks involving scene-dependent anomalies.

## 2 Related Work

### 2.1 Video Anomaly Detection

Traditional VAD methods mostly rely on self-supervised training using normal data [8, 33, 40, 45, 46] or adopt video-level labels [35, 4, 41] with multiple instance learning [28, 20]. While effective in constrained settings, they struggle with generalization due to limited training data. Recent works [37, 42, 39, 29] introduce LVLMs into VAD to leverage the capabilities acquired through large-scale multimodal pretraining. Early efforts explored training-free use of LVLMs by prompting them to summarize abnormal patterns for detection [42, 39]. Subsequent studies [6, 29] leverage LVLMs to detect anomalies in the form of video question answering. Zhang et al. [44] enhance anomaly detection performance by fine-tuning pre-trained LVLMs with supervision on VAD datasets. Despite the impressive performance of LVLMs, whether they truly ground their predictions in visual evidence remains unclear. In this work, we reveal a hallucination phenomenon rooted in LVLMs' reliance on visual-textual co-occurrence patterns, which causes the model to be misled by frequent but irrelevant visual instances. Furthermore, we propose a DPO-based training framework to mitigate this issue by guiding the model toward visually grounded reasoning.

### 2.2 Mitigating Hallucination in Large Vision-Language Models

Hallucination, a prevalent issue in LVLMs, refers to the generation of outputs that are not grounded in the input [47, 9, 12, 3, 21]. To mitigate hallucinations, various methods have been explored, most of which focus on improving decoding strategies [13, 5, 38]. For instance, Leng et al. [13] introduce visual contrastive decoding, which corrects hallucinated outputs by comparing the model's responses to original and perturbed visual inputs. Chen et al. [5] highlight the importance of incorporating both local and global visual context, which integrates an external grounding module during decoding. Xing et al. [38] perform attention reallocation during decoding to guide the model's focus toward visual information. In addition, some methods [10, 16, 48] address hallucination by performing robust instruction tuning on curated datasets. Despite these advances, hallucination in VAD remains underexplored. In this paper, we investigate this phenomenon from a co-occurrence perspective and introduce VAD-DPO, a training framework designed to mitigate hallucinations. To the best of our knowledge, this is the first systematic study of hallucination in the VAD setting.

## 3 Diagnosing Hallucinations in VAD: A Co-occurrence Perspective

**Task Formulation.** Given a video clip and a textual prompt, the objective of LVLM-based VAD [6, 44] is to determine whether the clip contains anomalous content. Specifically, the input to the model consists of a video sequence $\mathbf{v}$ and a question $\mathbf{q}$, typically phrased in natural language (e.g., "Is there anything abnormal in this video?"). The LVLM processes the visual and textual inputs and outputs a response $\mathbf{r}$, which may take the form of a binary decision ("Yes" or "No") or an open-ended caption. Formally, the LVLM-based VAD task is defined as:

$$\mathbf{r} = \mathtt{LVLM}(\mathbf{v}, \mathbf{q}), \tag{1}$$

where $\mathbf{v} = \{f_1, f_2, \ldots, f_T\}$ denotes a sequence of $T$ video frames, and $\mathbf{q}$ is the anomaly-related query. The model's output $\mathbf{r}$ reflects its internal judgment on whether the video contains an anomaly. To assess whether these models perform scene-aware anomaly reasoning or rely on spurious statistical shortcuts, we examine their behavior from a perspective of co-occurrence.

### 3.1 Does Hallucination Exist? Evidence from Object-Level Co-occurrence

In language modeling, co-occurrence refers to the frequency with which tokens appear together in training corpora. Extending this notion to vision-language settings, we hypothesize that anomaly judgments may be biased by visual instances (e.g., objects, scenes) that frequently co-occur with anomaly-related phrases (e.g., "fire," "robbery," "panic") during pretraining. Since LVLMs are not trained with explicit normal/anomalous labels, we approximate such biases by measuring alignment

Table 1: Object-level co-occurrence statistics between visual objects and abnormality/normality-related textual phrases. Top 8 objects are ranked by abnormal co-occurrence count; the last 2 are neutral visual objects added as counter-shortcut examples.

| Visual Object | Abn. Count | Norm. Count | Total Occur. | Abn./Norm. Ratio | Normalized Abn. Freq | Bias Score |
|---|---|---|---|---|---|---|
| gun | 812 | 34 | 846 | 23.9 | 0.960 | 5.24 |
| knife | 431 | 16 | 447 | 26.9 | 0.964 | 5.43 |
| fire | 688 | 51 | 739 | 13.5 | 0.931 | 4.40 |
| blood | 279 | 11 | 290 | 25.4 | 0.962 | 5.32 |
| smoke | 403 | 27 | 430 | 14.9 | 0.937 | 4.27 |
| ambulance | 209 | 19 | 228 | 11.0 | 0.917 | 3.54 |
| police car | 201 | 26 | 227 | 7.73 | 0.886 | 3.16 |
| syringe | 165 | 13 | 178 | 12.7 | 0.927 | 3.80 |
| tree | 17 | 62 | 79 | 0.27 | 0.215 | -1.87 |
| chef knife | 23 | 88 | 111 | 0.26 | 0.207 | -1.93 |

Table 2: False positive rates (FPR, %) on Probe Set 1 and false negative rates (FNR, %) on Probe Set 2. Higher values indicate stronger shortcut reliance.

| Model | Param. | Probe Set 1 (FPR) | | | | Avg. | Probe Set 2 (FNR) |
|---|---|---|---|---|---|---|---|
| | | Fire | Gun | Blood | Knife | | |
| Qwen2.5-VL [1] | 7B | 80.0 | 65.0 | 67.5 | 57.5 | 67.5 | 77.5 |
| Qwen2.5-VL [1] | 32B | **67.5** | **47.5** | **60.0** | 50.0 | **56.3** | 60.0 |
| LLaVA-1.5 [17] | 7B | 85.0 | 80.0 | 77.5 | 72.5 | 78.8 | 82.5 |
| LLaVA-1.5 [17] | 13B | 75.0 | 62.5 | 65.0 | 60.0 | 65.6 | 75.0 |
| LLaVA-NeXT-Video [18] | 7B | 82.5 | 70.0 | 67.5 | 55.0 | 68.8 | 72.5 |
| LLaVA-NeXT-Video [18] | 34B | **67.5** | 52.5 | 62.5 | **47.5** | 57.5 | **57.5** |
| InternVL3 [49] | 8B | 75.0 | 72.5 | 65.0 | 60.0 | 68.1 | 80.0 |

between visual content and anomaly-related or normality-related texts. As a first step, we examine whether the presence or absence of specific objects, which are objectively observable factors, alters the model's prediction. This perspective allows us to probe whether the model's output reflects scene-aware understanding or merely shallow vision-text statistical associations.

**Co-occurrence Statistics.** We conduct a statistical analysis on the LAION-CC-SBU dataset [17], which contains 558K image-text pairs and serves as part of the pretraining corpus for LLaVA [17, 18]. We begin by using GPT [24] to generate 3k candidate anomaly-related and normality-related textual phrases. These phrases are manually reviewed to ensure semantic clarity and accurate categorization. For each associated image, we apply YOLOv5 to detect objects and extract corresponding visual instances. We then compute co-occurrence statistics by counting how frequently each detected object appears in images whose captions contain either anomaly-related or normality-related phrases.

Table 1 presents the object-level co-occurrence statistics for the top 8 visual objects with the highest abnormal phrase co-occurrence, along with 2 counter-shortcut examples (e.g., *chef knife*) that often appear in normal contexts. We report the abnormal and normal co-occurrence counts, total occurrences, the abnormal-to-normal ratio, and the normalized abnormal frequency (NAF). To further quantify the bias, we compute a *Bias Score* using log-ratio scaling. The two metrics are defined as:

$$\text{NAF} = \frac{\text{Abn. Count}}{\text{Abn. Count} + \text{Norm. Count}}, \quad \text{Bias Score} = \log_2\left(\frac{\text{Abn. Count} + 1}{\text{Norm. Count} + 1}\right) \quad (2)$$

Higher values for both metrics indicate stronger co-occurrence bias toward abnormal semantics.

**Probe Setting.** To assess whether LVLMs rely on shortcut correlations, we construct two diagnostic probe sets. **Probe Set 1** targets false positives by inserting anomalous-looking objects—*fire*, *gun*, *blood*, and *knife*—into semantically normal scenes, with 40 samples per object selected based on Table 1. **Probe Set 2** targets false negatives by placing normal-looking objects in semantically abnormal contexts, also comprising 40 samples. In total, we generate 200 samples using frames

Table 3: Co-occurrence statistics of visual patterns with anomaly- and normality-related textual phrases. Bias scores reflect preference toward anomaly.

| Level | Visual Pattern | Abn. Count | Norm. Count | Abn./Norm. Ratio | Bias Score |
|---|---|---|---|---|---|
| Inter-action | person driving a car | 143 | 312 | 0.46 | -0.79 |
| | person lying on sidewalk | 298 | 19 | 15.7 | 3.97 |
| | person hitting another | 377 | 8 | 47.1 | 5.89 |
| | person walking a dog | 21 | 231 | 0.09 | -3.46 |
| Temporal | people running | 321 | 74 | 4.34 | 2.53 |
| | person jumping | 114 | 27 | 4.22 | 2.48 |
| | person standing still | 9 | 198 | 0.05 | -4.46 |
| | people walking slowly | 35 | 310 | 0.11 | -3.15 |

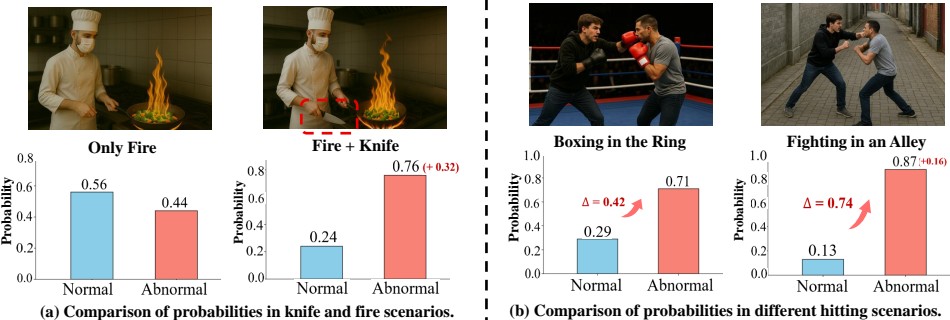

Figure 2: Effect of object combination and interaction-level co-occurrence on predicted anomaly probabilities. The presence of frequently co-occurring objects or interactions leads to high anomaly scores, even in semantically normal scenes, highlighting the influence of co-occurrence bias.

from VAD benchmarks [2, 44, 50] and Wan 2.1 [31], ensuring temporal consistency to avoid visual artifacts. We report false positive and false negative rates across multiple LVLMs.

**Observations.** As shown in Table 2, all models exhibit substantial false positive rates on the shortcut probe set, with even the strongest models, such as Qwen2.5-VL (32B), misclassifying over 50% of anomalous-looking-but-normal samples. Models like LLaVA 1.5 (7B) and InternVL3 (8B) reach false positive rates as high as 78.8% and 68.1%, respectively, indicating a strong reliance on object-level co-occurrence shortcuts. While scaling model size provides moderate improvements (e.g., Qwen2.5-VL 32B vs. 7B), the bias remains persistent across architectures. Notably, only LLaVA-NeXT-Video (34B) achieves a false negative rate below 60%, further confirming that models struggle to perform scene-aware reasoning about anomalies.

**Insight.** The results reveal a hallucination pattern: LVLMs tend to detect anomalies based on superficial vision-text co-occurrence learned during pretraining, rather than performing scene-aware reasoning. This shortcut behavior persists across architectures and model scales.

## 3.2 Beyond Object-Level: Analyzing Multi-Level Co-occurrence in VAD Hallucination

The previous analysis shows that LVLMs hallucinate anomalies based on object-level co-occurrence. Building on this finding, we further examine whether such shortcut reliance extends to more complex semantic structures. Specifically, we investigate whether hallucinations persist at higher semantic levels, including object combinations, interaction patterns, and temporal dynamics.

**Combination-Level.** To examine whether hallucination extends beyond individual objects, we construct combination-level probes by pairing object categories that exhibit strong anomaly-related co-occurrence, as identified in Table 1. Specifically, we conduct experiments on Qwen2.5-VL (7B) by inserting *knife* into the previously constructed *fire* samples from Probe Set 1, while ensuring the overall scene semantics remain normal. As shown in Figure 2(a), the predicted anomaly probability

significantly increases after inserting *knife*, even though the scene remains a semantically normal kitchen. This confirms that hallucination is not limited to isolated visual triggers but can be further reinforced by frequently co-occurring object combinations.

**Interaction-Level.** Beyond static object presence, we examine whether hallucinations can also be triggered by specific interaction patterns between objects or people. We focus on visually ambiguous interactions, such as *driving*, *hitting*, or *walking*, which are highly context-dependent and can appear either normal or abnormal depending on the scene. To quantify such interaction-level co-occurrence, we use GLIP [15] to detect interactions in videos associated with anomaly-related or normality-related text descriptions. The resulting co-occurrence statistics are summarized in Table 3.

Based on Table 3, we select the highly biased interaction pattern *person hitting another* and place it in two different contexts: a street alley to represent a semantically abnormal scene, and a boxing ring to represent a normal one. As shown in Figure 2(b), the model assigns significantly higher anomaly scores to both the street and boxing ring scenarios. This demonstrates that interaction-level co-occurrence can also interfere with the model's judgment, leading to hallucinations.

**Temporal-Level.** We further investigate whether hallucinations can be caused by temporal patterns that frequently co-occur with anomaly-related descriptions. To quantify this, we use Video-MAE [30] to extract motion categories from videos and compute their co-occurrence with anomaly- and normality-related texts. The resulting statistics are shown in Table 3.

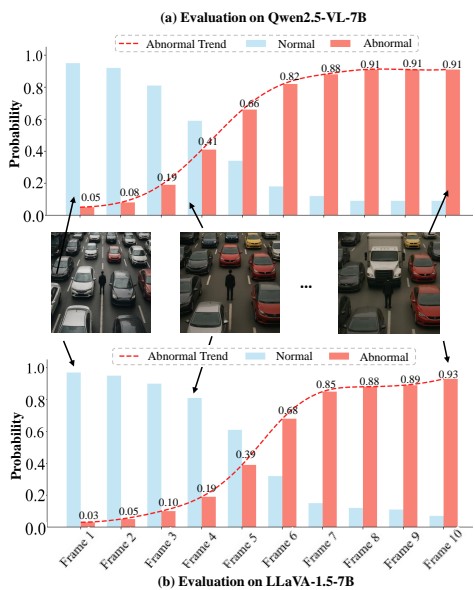

To probe whether temporal co-occurrence leads to hallucination, we construct an abnormal scene where a person stands still in the middle of a busy road. This visual pattern frequently co-occurs with normality-related text, yet it constitutes a clear semantic anomaly in this context. Unlike previous cases, LVLMs such as Qwen2.5-VL [1] and LLaVA-1.5 [17] are not misled by the co-occurrence and identify the scene as abnormal.

To understand this behavior, we visualize frame-level anomaly probabilities. As shown in Figure 3, the models initially predict normality but gradually detect the anomaly as the sequence unfolds. This indicates that temporal progression provides additional semantics for scene-aware reasoning.

Figure 3: Frame-wise anomaly probabilities on a temporal probe.

**Insight.** Hallucinations in VAD extend beyond object-level to co-occurrence patterns involving object combinations, interactions, and temporal dynamics. While combination and interaction patterns often reinforce false positives, temporal information can both mislead and correct model predictions.

## 4 Proposed Method: Direct Preference Optimization for VAD

To address the issue revealed in Section 3, we propose **VAD-DPO**, a Direct Preference Optimization approach for VAD. VAD-DPO reformulates detection as a preference alignment task, encouraging the model to favor outputs consistent with semantic correctness, thereby guiding it to reason based on scene-level semantics rather than relying on superficial statistical shortcuts.

### 4.1 Preliminary

Direct Preference Optimization (DPO) [26] aligns model behavior with human preferences by directly optimizing over preference pairs without reinforcement learning or external reward models. Given a multi-modal input $(x, v)$ and two candidate responses $y_w$ (preferred) and $y_l$ (less preferred), the model $\pi_\theta$ defines a conditional distribution $\pi_\theta(y|x, v)$. DPO encourages the model to increase the relative likelihood of the preferred output while regularizing divergence from a reference model $\pi_{\text{ref}}$,

initialized from the same checkpoint. The training objective is derived from the Bradley-Terry model:

$$\mathcal{L}_{\text{DPO}} = -\log \sigma \left( \beta \cdot \log \frac{\pi_\theta(y_w \mid x, v)}{\pi_{\text{ref}}(y_w \mid x, v)} - \beta \cdot \log \frac{\pi_\theta(y_l \mid x, v)}{\pi_{\text{ref}}(y_l \mid x, v)} \right), \qquad (3)$$

where $\sigma(\cdot)$ is the sigmoid function and $\beta$ controls the sharpness of preference alignment.

## 4.2   Optimization Objective for VAD

Building on recent advances in multimodal preference optimization [16, 48, 11], we formulate anomaly detection as a direct preference optimization problem. To explicitly break shortcut reliance, we construct *counter-example preference pairs* from existing datasets [50, 44] and further synthesize additional pairs with Wan2.1 [31], where clips are visually similar but semantically contrasting. For instance, both "a person hitting another in a boxing ring" and "a person hitting another in a street alley" involve the same high-frequency interaction pattern (hitting), which co-occurs with anomaly-related cues. However, only the latter should be judged anomalous. By training the model to prefer the semantically correct response $y_w$ over the incorrect $y_l$, VAD-DPO enforces reasoning grounded in scene semantics rather than superficial correlations.

Formally, let $(x, v)$ denote the textual prompt and video input, and $(y_w, y_l)$ be two candidate outputs. Following the DPO objective, the VAD-specific loss is:

$$\mathcal{L}_{\text{VAD-DPO}} = -\log \sigma \left( \beta \cdot \log \frac{\pi_\theta(y_w \mid x, v)}{\pi_{\text{ref}}(y_w \mid x, v)} - \beta \cdot \log \frac{\pi_\theta(y_l \mid x, v)}{\pi_{\text{ref}}(y_l \mid x, v)} \right). \qquad (4)$$

To further stabilize training and preserve strong preference for semantically valid clips, we adopt an anchored objective [32]:

$$\mathcal{L}_{\text{Anc}} = -\log \sigma \left( \beta \cdot \log \frac{\pi_\theta(y_w \mid x, v)}{\pi_{\text{ref}}(y_w \mid x, v)} \right). \qquad (5)$$

The final loss combines contrastive and anchored terms:

$$\mathcal{L}_{\text{total}} = \mathcal{L}_{\text{VAD-DPO}} + \gamma \cdot \mathcal{L}_{\text{Anc}}, \qquad (6)$$

where $\gamma$ balances the contribution of anchoring. Through counter-example preference pairs, VAD-DPO directly targets the co-occurrence shortcut issue, compelling the model to distinguish visually similar but semantically distinct contexts. This preference-based supervision shifts the model from surface-level correlations toward robust reasoning over scene semantics, thereby mitigating shortcut-driven hallucinations in LVLM-based VAD.

## 4.3   LVLM-based Video Anomaly Detection

During inference, we adopt the evaluation protocol of LAVAD [42], which consists of two stages. (1) **Frame Sampling and Scoring:** For each video segment, we uniformly sample 8 frames and pair each with a scoring-style prompt, following [42]. For example, we ask the LVLM: *"If you were a law enforcement agency, how would you rate the scene on a scale from 0 to 1, where 0 denotes a standard scene and 1 indicates suspicious activities?"* The model directly outputs a numerical score, which is taken as the anomaly score of the frame. (2) **Score Refinement and Assignment:** The sparse frame-level scores are refined through a weighted aggregation mechanism. Specifically, we exploit temporal attention weights from the decoding process to propagate scores from sampled frames to their neighbors, producing a dense per-frame anomaly score sequence across the entire video.

# 5   Experiments

## 5.1   Experiment Setup

**Datasets.** We evaluate our method on six real-world surveillance datasets commonly used in VAD: ShanghaiTech [19], UCF-Crime [28], XD-Violence [35], NWPU Campus [2], MSAD [50], and HIVAU-70K [44]. ShanghaiTech [19] features 13 surveillance views from a campus scene. UCF-Crime [28] contains over 1,900 hours of videos across 13 crime categories. XD-Violence [35]

Table 4: Comparison of frame-level AUC (%) with state-of-the-art LVLMs and VAD methods on ShanghaiTech, UCF-Crime, XD-Violence, NWPU Campus, and MSAD.

| Method | Params | SHTech | UCF-Crime | XD-Violence | NWPU Campus | MSAD |
|---|---|---|---|---|---|---|
| Qwen2.5-VL [1] | 7B | 79.4 | 78.8 | 83.2 | 71.9 | 75.9 |
| LLava-1.5 [17] | 13B | 76.3 | 72.8 | 79.6 | 70.3 | 75.1 |
| LLaVA-NeXT-Video [18] | 7B | 78.8 | 75.4 | 81.5 | 72.6 | 78.6 |
| LAVAD [42] | 13B | 81.8 | 80.3 | 85.4 | 71.1 | 79.4 |
| A-Guardian [6] | 7B | 80.9 | 76.1 | 85.0 | 72.3 | 80.9 |
| AnomalyRuler [39] | 7B | 84.6 | 78.6 | 83.1 | 69.9 | 81.1 |
| Hawk [29] | 7B | 80.8 | 81.9 | 80.7 | 72.5 | 78.9 |
| Holmes-VAU [44] | 2B | 85.2 | 84.5 | 86.3 | 70.8 | 81.6 |
| **VAD-DPO (Ours)** | **2B** | **87.2** | **86.2** | **88.5** | **79.1** | **85.4** |

Table 5: False positive rates (FPR, %) on Probe Set 1 and false negative rates (FNR, %) on Probe Set 2. Higher values indicate stronger shortcut reliance.

| Model | Param. | Probe Set 1 (FPR) | | | | Avg. | Probe Set 2 (FNR) |
|---|---|---|---|---|---|---|---|
| | | Fire | Gun | Blood | Knife | | |
| LAVAD [42] | 13B | 85.0 | 82.5 | 80.0 | 87.5 | 82.5 | 82.5 |
| A-Guardian [6] | 7B | 77.5 | 80.0 | 85.0 | 85.0 | 81.9 | 80.0 |
| AnomalyRuler [39] | 7B | 82.5 | 87.5 | 90.0 | 80.0 | 81.9 | 82.5 |
| Hawk [29] | 7B | 90.0 | 80.0 | 77.5 | 72.5 | 85.0 | 77.5 |
| Homles-VAU [44] | 2B | 75.0 | 80.0 | 72.5 | 72.5 | 75.0 | 80.0 |
| **VAD-DPO(Ours)** | **2B** | **15.0** | **12.5** | **17.5** | **7.5** | **13.1** | **12.5** |

includes 4,754 video clips from movies, online sources, and CCTV, focusing on violence detection. NWPU Campus [2] provides 43 scenes, with a majority involving scene-dependent anomalies. MSAD [50] comprises data from 14 scenes and 500 viewpoints, offering large-scale scene diversity. HIVAU-70K [44], built upon UCF-Crime and XD-Violence, is tailored for LVLM-based VAD with multi-granularity segmentation and rich textual descriptions.

**Evaluation Metrics.** To comprehensively evaluate LVLMs in VAD, we consider both detection performance and reasoning ability. For detection, we report Area Under the ROC Curve (AUC) as the main metric. In addition, we report false positive rate (FPR) on Probe Set 1 and false negative rate (FNR) on Probe Set 2.

**Implementation Details.** We set the learning rate to 1e-4 and adopt a cosine learning rate scheduler with a warm-up ratio of 0.05. The default value of $\gamma$ is set to 1 in Equation 6. All models are trained for a single epoch.

## 5.2 Main Results

While most existing LVLM-based VAD methods [42, 6, 39, 29] operate in a training-free or zero-shot setting, Holmes-VAU [44] is the only work that explores supervised fine-tuning of an LVLM. Accordingly, we implement VAD-DPO on Qwen2-VL (2B) [34], aligning with the parameter scale of Holmes-VAU to ensure a fair comparison.

**Detection Performance.** We evaluate the frame-level detection performance of VAD-DPO on five benchmark datasets and compare it with state-of-the-art LVLM-based VAD methods as well as representative open-source LVLMs. For methods that did not report frame-level AUC on these benchmarks, we reproduce the results using their publicly released weights and code, without any modification. As shown in Table 4, VAD-DPO achieves the highest AUC scores across all benchmark datasets, outperforming both 7B and 13B LVLMs despite using only 2B-scale parameters. Compared to Holmes-VAU [44], VAD-DPO achieves at least a 2% improvement in AUC across all benchmarks, with a notable gain of 8.3% on NWPU Campus. The NWPU Campus dataset contains scene-dependent anomalies, which require the model to understand contextual semantics. The significant improvement on this benchmark demonstrates that VAD-DPO effectively guides the model to perform scene-aware reasoning, rather than relying on superficial statistical shortcuts.

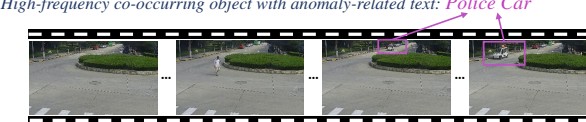

| | |
|---|---|
| **System Prompt:** Determine whether an abnormal event occurs in the given video. | *High-frequency co-occurring object with anomaly-related text: Police Car* |
| **Scene Semantic:** A normal traffic intersection where a police car is on routine patrol. | |

---

*Prediction of Qwen2.5-VL-7B:*

Yes, a police car is approaching in the scene, indicating that an abnormal event is occurring.

---

*Prediction of LLaVA-NeXT-Video-7B:*

Yes, a police car appears at the intersection, which may indicate that an abnormal event has occurred.

---

*Prediction of LLaVA-1.5-13B:*

Yes, the appearance of a police car at the intersection may indicate a potential anomaly.

---

*Prediction of Ours:*

No, the pedestrian is walking through the intersection normally, and a police car is slowly approaching. No anomaly is detected in the scene.

Figure 4: Qualitative comparison illustrating hallucination in LVLM-based VAD. While Qwen2.5-VL-7B, LLaVA-NeXT-Video-7B, and LLaVA-1.5-13B incorrectly predict an anomaly due to the presence of a high-frequency co-occurring object (police car), our method accurately identifies the scene as normal based on contextual understanding.

**Performance on Shortcut Probes.** Table 5 extends the results in Table 2 by including both domain-specific expert models [44, 29], which are fine-tuned on VAD datasets, and training-free LVLM-based methods [42, 6, 39]. All results are obtained using released code and checkpoints, without retraining or manual tuning. We use a fixed threshold of 0.5 across all models to compute false positive and false negative rates, ensuring fair comparison despite differences in output score distributions.

Although the expert models are trained specifically for VAD, they still exhibit high hallucination rates. We attribute this to the scarcity of true anomaly samples in existing datasets, which biases supervised fine-tuning toward spurious co-occurrence patterns. In contrast, VAD-DPO reduces the average false positive rate from over 75% in expert models to only 13.1%, an absolute reduction of more than 80%, highlighting its effectiveness in overcoming shortcut reliance. Training-free methods [42, 6, 39] also suffer from severe hallucinations, which is expected given their dependence on earlier-generation LVLMs. For example, LAVAD [42] relies on BLIP2 with LLaMA-2-13B, MMeval uses Vicuna-7B, and AnomalyRuler employs Mistral-7B-Instruct-v0.2. These models lack sufficient multimodal reasoning capability and thus perform consistently worse than our approach.

**Qualitative Results.** Figure 4 illustrates the superior qualitative performance of our proposed VAD-DPO in detection and reasoning. Compared to other baselines, VAD-DPO accurately understands and focuses on the anomalous events within the video. In contrast, other LVLMs are influenced by statistical correlations learned during pre-training, which leads them to generate descriptions centered around frequently co-occurring patterns.

### 5.3 Ablation Study and Analysis

**Effect of Preference Alignment.** To evaluate the effectiveness of preference alignment in VAD-DPO, we conduct ablation studies on the NWPU Campus dataset. Using the pre-trained Qwen2-VL (2B) as the baseline, we compare the effects of different training strategies, including instruction tuning, Proximal Policy Optimization (PPO) [27], standard DPO [26], and VAD-DPO. All methods are trained on the same data from HIVAU-70K [44] for a fair comparison.

Table 6: AUC (%) on NWPU Campus under different training paradigms.

| Training Paradigm | AUC |
|---|---|
| Instruction Tuning [44] | 69.2 |
| PPO [27] | 72.3 |
| DPO [26] | 73.7 |
| VAD-DPO (Ours) | **79.1** |

As shown in Table 6, VAD-DPO significantly outperforms all other training paradigms, achieving an AUC of 79.1%. **Instruction tuning** performs worst (69.2%) due to its lack of guidance for contextual reasoning. **PPO** improves slightly (72.3%) but suffers from the difficulty of defining effective reward functions. **Standard DPO** achieves 73.7% but fails to explicitly guide the model to understand why

the preferred response aligns with the input video or why the rejected response should be discouraged. In contrast, **VAD-DPO** directly guides the model to suppress preferences for high-frequency patterns when they are contextually irrelevant, effectively mitigating reliance on statistical shortcuts.

**Effect of** $\gamma$. As shown in Figure 5, we investigate the impact of different $\gamma$ values on the performance of VAD-DPO. Across four benchmarks—UCF-Crime, XD-Violence, NWPU Campus, and MSAD—we observe that smaller $\gamma$ values significantly improve detection performance, indicating effective mitigation of hallucinations. However, as $\gamma$ increases, performance gradually degrades due to excessive suppression of co-occurrence patterns, which impairs the model's reasoning ability. To balance these effects, we set $\gamma$ to 1 by default, aiming to reduce the model's reliance on statistical shortcuts during reasoning without compromising anomaly detection performance.

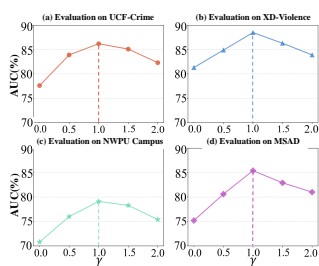

Figure 5: Ablation studies on $\gamma$.

**Generalization of VAD-DPO.** We evaluate the resulting VAD-DPO model on four general-purpose video understanding benchmarks, including egocentric action recognition, schematic reasoning, and video question answering. As shown in Table 7, VAD-DPO achieves consistent improvements over the original Qwen2-VL-2B. This suggests that preference tuning not only preserves generalization ability, but may also

Table 7: Generalization of VAD-DPO.

| Benchmark | Qwen2-VL | Ours |
|---|---|---|
| MVBench [14] | 63.2 | **63.6** |
| PerceptionTest [25] | 53.9 | **54.5** |
| EgoSchema [23] | 54.9 | **55.1** |
| Video MME [7] | 60.4 | **61.2** |

enhance it by strengthening semantic grounding. Overall, the results demonstrate that VAD-DPO maintains robust generalization even when trained on a relatively small but carefully constructed set of preference pairs.

## 6   Conclusion

In this paper, we investigate the behavior of LVLMs in video anomaly detection from the perspective of co-occurrence between visual instances and textual phrases. By analyzing pre-training datasets and conducting probe experiments, we reveal a hallucination phenomenon in LVLM-based VAD: LVLMs tend to rely on statistical shortcuts learned during pre-training rather than performing scene-aware contextual reasoning when identifying anomalies. Further controlled studies demonstrate that co-occurrence at the combination and interaction levels exacerbates this issue. To address this problem, we propose VAD-DPO, which formulates VAD as a pairwise preference optimization task. Specifically, the model is trained on visually similar video clips with contrasting anomaly labels to encourage alignment with semantic consistency rather than co-occurrence shortcuts. Extensive experiments across six benchmark datasets validate the effectiveness of our method.

**Future Work.**   Recent advances such as GRPO provide promising directions for the future development of LVLMs, we believe there remains significant potential in adapting GRPO-style optimization to video anomaly detection. In particular, future work may explore: (1) tailoring group-wise ranking strategies in GRPO to better capture the semantic that are central to VAD; (2) integrating task-specific reward models, to stabilize training and prevent semantic drift; and (3) scaling GRPO with larger corpora to fully leverage its potential.

## Acknowledgements

This work was supported in part by the National Natural Science Foundation of China under Grants (62406039, 62321001, 62471055, U23B2001, 62101064, 62171057, 62071067), the High-Quality Development Project of the MIIT(2440STCZB2584), the Ministry of Education and China Mobile Joint Fund (MCM20200202, MCM20180101), the Project funded by China Postdoctoral Science Foundation (2023TQ0039, 2024M750257, GZC20230320), the Fundamental Research Funds for the Central Universities (2024PTB-004), the 2025 Education and Teaching Reform Project Funding at Beijing University of Posts and Telecommunications (2025YZ005).

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

# Appendix

In this appendix, we provide additional experimental results, implementation details, probe dataset construction methods, and further qualitative and quantitative analyses to support the claims made in the main paper. These materials are organized as follows:

▶ **Section A Supplementary Analyses and Technical Details**: We provide supplementary analyses and technical details, including probe design, co-occurrence statistics, implementation details, extended results, and more qualitative examples.

▶ **Section A.1 Diagnostic Probe Construction and Examples**: We detail the construction process of diagnostic probe datasets, including visual examples for both false positives and false negatives.

▶ **Section A.2 High-Level Co-occurrence Statistics and Analysis**: We present the high-level co-occurrence statistics across object combinations, interaction patterns, and temporal dynamics, along with quantitative evaluations that extend our analysis beyond the object level.

▶ **Section A.3 Training and Implementation Details**: We describe the training and implementation details of VAD-DPO, including model initialization, optimization settings, and computational resources used.

▶ **Section A.4 Additional Experimental Results**: We report additional experimental results, including reasoning performance and the performance of implementing VAD-DPO on InternVL3.

▶ **Section A.5 Comparisons with Methods for Mitigating Hallucination**: We further clarify our contribution, discuss hallucination in LVLMs, and compare with related work.

▶ **Section A.6 More Qualitative Examples**: We show more qualitative examples comparing VAD-DPO and baseline models under various scenarios, highlighting differences in reasoning behavior.

▶ **Section B Limitations**: We discuss the limitations of our current approach, including model scale and residual hallucination effects, and outline possible directions for future work.

▶ **Section C Broader Societal Impacts**: We discuss the broader societal impacts of our work, highlighting the potential benefits in safety-critical applications.

## A Supplementary Analyses and Technical Details

### A.1 Diagnostic Probe Construction and Examples

To evaluate whether LVLMs rely on co-occurrence-driven shortcuts rather than truly understanding scene semantics, we construct two diagnostic probe sets designed to induce controlled hallucination scenarios. This section complements the analysis in Section 3.1 by providing detailed descriptions of how these probe sets are constructed, their underlying rationale, and the evaluation protocol used to assess hallucination behavior across different LVLMs.

**Details of Co-occurrence Statistics.** As described in Section 3.1, we conduct this analysis using the LAION-CC-SBU dataset. We begin by using GPT to generate approximately 3k candidate textual phrases that are potentially associated with anomaly or normality. These phrases are manually reviewed to ensure semantic clarity and proper categorization. We then match these phrases against LAION captions to identify text-image pairs where a caption contains a candidate phrase. For each matched image, we apply YOLOv5 to detect objects and filter out low-confidence detections to ensure statistical robustness. This process allows us to compute co-occurrence statistics by measuring how frequently certain visual objects appear in images whose captions express either anomaly-related or normality-related concepts. In this way, the co-occurrence patterns are estimated from large-scale real-world data with careful semantic control.

**GPT-generated Textual Phrases.** To construct our probing datasets, we use GPT-4 to generate a diverse set of 3k candidate textual phrases related to anomaly and normality. We prompt the model with instructions such as: Generate a list of concise phrases that describe either abnormal or normal events in video scenes. Some example outputs include:

- Anomaly-related: "a person holding a knife", "fire spreading rapidly", "man lying motionless on the ground".

- Normality-related: "people walking across a street", "children playing in a park", "car driving along a road".

We manually review and filter these phrases to ensure semantic clarity and diversity.

**Probe Set 1: False Positive Induction.** This set aims to test whether models incorrectly predict anomalies when anomalous-looking objects are introduced into semantically normal scenes. Based on the object-level co-occurrence statistics in Table 1, we identify four visual objects, *fire*, *gun*, *blood*, and *knife*, that exhibit strong abnormality bias. For each object, we manually select 40 base scenes from VAD datasets where the original context is clearly normal. We then use image editing or video generation techniques [31] to insert the target object in a visually realistic way, ensuring that the modified frame still maintains normal semantics overall. These modified clips are used to test the model's hallucination susceptibility.

**Probe Set 2: False Negative Induction.** To evaluate whether LVLMs overlook semantic anomalies when scenes contain superficially benign objects, we design a second probe set. We select 40 semantically abnormal situations where abnormality arises from scene context rather than object presence. Into these scenes, we insert neutral or high-frequency objects associated with normality (e.g., *tree*, *chef knife*) that may distract or bias the model toward false negatives. Examples include inserting a peaceful-looking object into a street fight, or placing a daily-use item in a chaotic environment.

**Dataset Sources and Consistency.** The probe videos are derived from two sources. First, we select representative examples from three real-world datasets—NWPU Campus [2], HIVAU-70K [44], and MSAD [50]—which serve as the base scenes. Second, we construct edited versions of these samples using Wan2.1 [31], enabling controlled insertion of objects or contextual modifications to induce hallucination. We ensure that temporal consistency is preserved after object insertion by verifying motion cues and avoiding visual artifacts. Each probe video contains a sequence of frames with minimal modification outside the target region to isolate the effect of co-occurrence-driven bias.

**Human Validation Protocol.** Our goal in constructing probe samples is to simulate scenes where the most plausible and commonsense interpretation is normal, despite the presence of anomaly-related objects. For example, a video depicting a man barbecuing in a backyard may include a "fire" object, yet the overall scene is semantically normal. To ensure the normality, we applied a strict human validation protocol:

- Manual Screening: All videos are reviewed by multiple annotators to ensure that the inserted object does not introduce unintended anomaly cues beyond its visual presence.
- Exclusion of Ambiguous Cases: We filter out videos in which the anomaly-related object could reasonably support multiple semantic interpretations (e.g., police cars in tense crowd scenes).

While we acknowledge a degree of subjectivity is inevitable, this validation protocol is designed to minimize such ambiguity.

**Evaluation Protocol.** We evaluate multiple LVLMs of varying sizes and architectures (Qwen2.5-VL, LLaVA-1.5, LLaVA-NeXT-Video, InternVL3) on both probe sets. For Probe Set 1, we report false positive rates (FPR), indicating hallucination susceptibility when no actual anomaly exists. For Probe Set 2, we report false negative rates (FNR), measuring the model's failure to detect context-based anomalies. Results are summarized in Table 2, showing that models consistently exhibit higher error rates in these carefully constructed edge cases.

**Visual Examples.** Figure A.1 provides representative examples from both probe sets, including four cases from Probe Set 1 and two from Probe Set 2. These examples illustrate how high-bias or contextually misleading objects are inserted into real scenes to construct controlled test cases. To protect privacy, all visible faces in the frames have been masked. We acknowledge that simply applying masks may not be sufficient to fully eliminate potential ethical and privacy concerns. In future releases of our probe sets, we plan to adopt full-face blurring or visual obfuscation techniques to further enhance privacy protection.

## A.2   High-Level Co-occurrence Statistics and Analysis

**Overview.** This section provides implementation details and quantitative analysis of high-level co-occurrence patterns that may lead to hallucinations in LVLM-based VAD systems. Specifically,

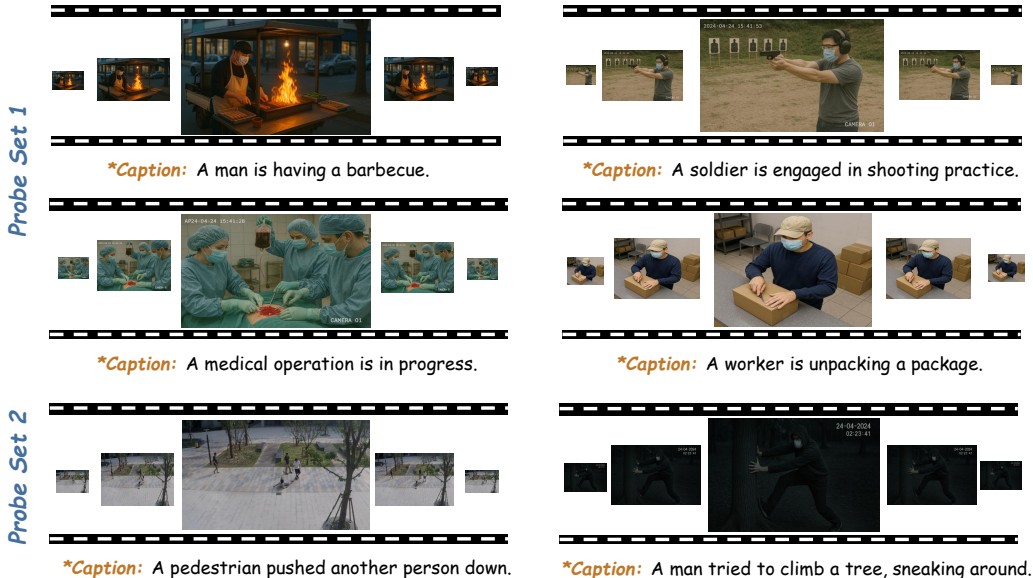

Figure A.1: Representative examples from the diagnostic probe sets. The top row shows four samples from Probe Set 1, where high-bias objects (e.g., *fire*, *gun*) are inserted into semantically normal scenes to induce false positives. The bottom row shows two samples from Probe Set 2, where contextually abnormal scenes contain visually benign elements that may mislead the model toward false negatives. All visible faces have been masked to preserve privacy.

we elaborate on the statistical construction of object combinations, interaction patterns, and temporal dynamics, and report the corresponding anomaly prediction behaviors.

**Object Combination Co-occurrence.** We identify object pairs with high anomaly-related co-occurrence frequency based on conditional probabilities computed from the HIVAU-70K dataset. Among them, *knife* and *fire* form a frequently co-occurring pair in anomalous contexts. To test whether their combination induces hallucination, we insert *knife* into *fire* scenes (originally labeled as normal) while keeping the scene semantics unchanged (e.g., a kitchen). Quantitative results show that adding *knife* increases the predicted anomaly probability significantly, despite the absence of actual anomalies.

**Interaction-Level Co-occurrence.** We use GLIP [15] to extract subject-verb-object interactions from video-text pairs and compute their anomaly association via normalized co-occurrence counts. The pattern *person hitting another* is selected due to its strong anomaly correlation. To isolate the effect of context, we embed the same interaction into two settings: a street alley and a boxing ring. In both, models incorrectly predict high anomaly scores, confirming interaction-level hallucination even in semantically normal contexts.

**Temporal Dynamics Co-occurrence.** Motion patterns are extracted using VideoMAE [30] and aligned with anomaly/normality tags through co-occurrence analysis. The motion pattern *standing still* is often associated with normality in the training data. We construct a probe where a person remains stationary in the middle of a busy road. While the scene is clearly anomalous, we observe that some models initially assign low anomaly scores, which only rise after several frames, indicating delayed recognition due to misleading temporal priors.

**Summary.** These results demonstrate that hallucination in LVLMs extends beyond object-level triggers to include more complex semantic structures. Object combinations, interaction patterns, and temporal motion—when biased by training co-occurrence—can each mislead the model into incorrect anomaly predictions, despite semantically normal visual input.

## A.3 Training and Implementation Details

We provide additional details regarding the training setup and implementation of VAD-DPO.

**Model Initialization.** All experiments are conducted using the Qwen2-VL [34] backbone with 2B parameters. The reference model $\pi_{\text{ref}}$ is initialized from the same checkpoint and frozen throughout training. We use the official HuggingFace implementation with minor modifications to support preference-based finetuning.

**Training Details.** We use mixed-precision training (fp16) and a batch size of 16 preference pairs per GPU. All models are trained for a single epoch using the AdamW optimizer with a learning rate of 1e-6 and cosine decay schedule (warm-up ratio 0.05). The alignment sharpness factor $\beta$ is set to 0.1 following DPO best practices [26], and $\gamma$ is set to 1 unless otherwise specified.

**Preference Pair Construction.** Preference pairs $(y_w, y_l)$ are constructed from HIVAU-70K [44], MSAD [50], and NWPU Campus [2]. Each pair consists of two video clips with similar visual content but differing semantics—only one clip is contextually anomalous. These clips are selected based on manual inspection of anomaly labels and scene descriptions, ensuring high semantic contrast with minimal visual confounds. In total, we curate 1,000 such preference pairs for training. To ensure both semantic contrast and visual alignment, most pairs are sourced in one of two ways: (1) from the same scene where different but visually similar behavior patterns (e.g., walking vs. running) occur under normal and anomalous contexts, or (2) from different real-world scenes that share the same surface-level activity (e.g., standing still) but differ in contextual interpretation. This construction strategy helps isolate the causal semantics of anomaly, making the preference signal more robust and interpretable.

**Training Time and Resources.** All training is performed on an internal cluster equipped with 2 NVIDIA A100 GPUs (80GB memory each), using distributed data parallelism via PyTorch's 'torch.distributed' module. VAD-DPO model is trained for one full epoch over the 1,000 constructed preference pairs, which corresponds to approximately 90,000 optimization steps (given pair-wise sampling and batching). The end-to-end training process takes around 16 GPU-hours.

To reduce data loading bottlenecks, all video clips are pre-extracted into frame sequences and cached on local SSDs. Training uses mixed precision (fp16) to improve memory efficiency and reduce training time by 30% compared to full precision.

Table A.1: Comparison of reasoning performance with state-of-the-art LVLMs on HIVAU-70k. BLEU, CIDEr, and ROUGE scores are reported at clip-level (C), event-level (E), and video-level (V).

| Method | Params | BLEU ↑ | | | CIDEr ↑ | | | ROUGE ↑ | | |
|---|---|---|---|---|---|---|---|---|---|---|
| | | C | E | V | C | E | V | C | E | V |
| Video-ChatGPT [22] | 7B | 0.152 | 0.068 | 0.066 | 0.033 | 0.011 | 0.013 | 0.153 | 0.048 | 0.079 |
| Video-LLaMA [43] | 7B | 0.151 | 0.079 | 0.104 | 0.024 | 0.014 | 0.017 | 0.156 | 0.067 | 0.090 |
| LLava-1.5 [17] | 7B | 0.316 | 0.097 | 0.189 | 0.115 | 0.018 | 0.028 | 0.176 | 0.077 | 0.091 |
| LLAVA-Next-Video [18] | 7B | 0.435 | 0.091 | 0.120 | 0.102 | 0.015 | 0.031 | 0.198 | 0.080 | 0.106 |
| Qwen2.5-VL [1] | 7B | 0.481 | 0.154 | 0.263 | 0.146 | 0.030 | 0.059 | 0.201 | 0.107 | 0.169 |
| Holmes-VAU [44] | 2B | 0.913 | 0.804 | 0.566 | 0.467 | 1.519 | 1.437 | 0.329 | 0.370 | 0.355 |
| **VAD-DPO**$_{\text{Holmes}}$ | **2B** | **0.923** | **0.881** | **0.645** | **0.688** | **1.984** | **1.832** | **0.561** | **0.596** | **0.573** |

## A.4 Additional Experimental Results

In Section 5, we evaluated the detection performance of VAD-DPO on mainstream benchmark datasets as well as its results on the constructed probe sets. In this appendix, we further report supplementary results, including reasoning performance on HIVAU-70K [44] and detection results on the XD-Violence dataset [35] using AP as the evaluation metric. Following [44], we assess the quality of generated descriptions on clip-, event-, and video-level test samples using BLEU, CIDEr, METEOR, and ROUGE scores.

**Reasoning Performance.** To ensure fair comparison and isolate the effect of fine-tuning, we implement VAD-DPO using Holmes-VAU [44] as the LVLM, denoted as VAD-DPO$_{\text{Holmes}}$. We compare VAD-DPO with state-of-the-art LVLMs, including Video-ChatGPT [22], Video-LLaMA [43], LLaVA-1.5 [17], LLaVA-NeXT-Video [18], Qwen2.5-VL [1], and Holmes-VAU [44]. As shown in Table A.1, VAD-DPO achieves the highest scores across all metrics, reflecting its ability to guide the model to reason based on scene semantics rather than relying on co-occurrence patterns, thereby generating descriptions that are more consistent with the visual context.

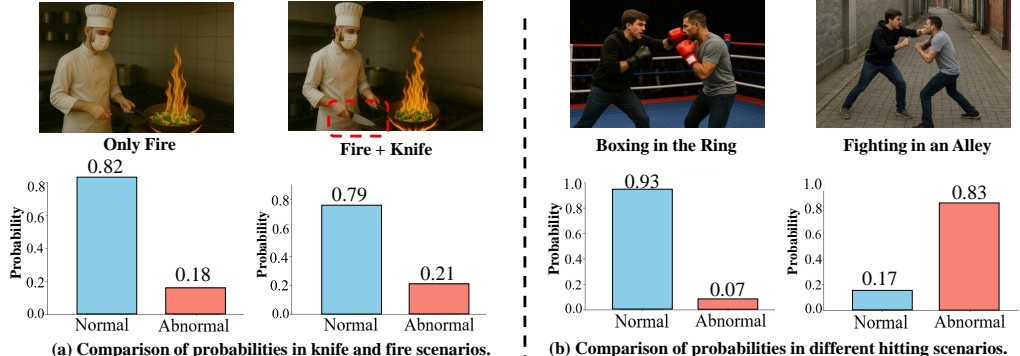

(a) Comparison of probabilities in knife and fire scenarios.  (b) Comparison of probabilities in different hitting scenarios.

Figure A.2: Qualitative comparison illustrating hallucination in LVLM-based VAD. While Qwen2.5-VL-7B, LLaVA-NeXT-Video-7B, LLaVA-1.5-13B, and InternVL-8B incorrectly predict an anomaly due to the presence of high-frequency co-occurring objects or patterns, our method accurately identifies the scene as normal by leveraging contextual understanding. Green boxes indicate correct predictions, red boxes indicate incorrect predictions, and purple highlights mark high-frequency co-occurring objects or patterns that may induce hallucination.

**Robustness to High-Level Co-occurrence.** Following the analysis in Section 3.2, we further evaluate VAD-DPO's robustness to high-level co-occurrence patterns. Specifically, we examine two challenging settings: (1) frequent co-occurrence of high-bias object *combinations*, and (2) identical *interactions* occurring in different scene contexts. For each case, we measure the anomaly probability predicted by VAD-DPO to assess whether the model is misled by co-occurrence patterns or correctly grounds its judgment in scene semantics.

As shown in Figure A.2, VAD-DPO consistently provides accurate anomaly predictions in both settings. It successfully avoids false alarms when object combinations appear in normal contexts, and correctly identifies anomalies when the same interaction becomes semantically abnormal in a different scene. These results validate VAD-DPO's robustness to high-level shortcut cues and its ability to make context-aware, semantically grounded predictions.

**Results on Different LVLMs.** To further evaluate our proposed approach, we apply VAD-DPO to InternVL3-1B [49]. We select InternVL3-1B for fine-tuning to maintain a parameter scale comparable to Qwen2-VL-2B. The results are summarized in Table A.2. Specifically, VAD-DPO improves the average frame-level AUC of InternVL3-1B on VAD

Table A.2: Performance of VAD-DPO on InternVL3-1B compared to baseline LVLMs

| LVLM | Param | NWPU AUC ↑ | Probe Set 1 FPR ↓ | Probe Set 2 FNR ↓ |
|------|-------|------|------|------|
| InternVL3 | 8B | 71.2 | 70.0 | 75.0 |
| InternVL3 | 1B | 69.3 | 75.0 | 80.0 |
| **VAD-DPO** (InternVL3) | 1B | **76.3** | **24.4** | **27.5** |

benchmark by more than 10%, while reducing hallucination-induced error rates from 75.0% to 24.4% on Probe Set 1 and from 80.0% to 27.5% on Probe Set 2. Despite having significantly fewer parameters, InternVL3-1B fine-tuned with VAD-DPO outperforms the off-the-shelf InternVL3-8B on both standard VAD benchmarks and hallucination-sensitive probe sets. These results demonstrate that our method is effective not only for Qwen-based models but also generalizes well to other LVLM architectures.

## A.5 Comparisons with Methods for Mitigating Hallucination

Co-occurrence has been identified as a cause of object hallucination in prior VLM studies. However, we argue that the nature of co-occurrence and its resulting failure mode in VAD are fundamentally different, which constitutes the core focus of our work. Object hallucination typically arises from object-context co-occurrence. In contrast, VAD hallucination stems from object-label co-occurrence: the model does not invent nonexistent objects but correctly detects real ones (e.g., a fire in the scene) and mistakenly classifies them as anomalous solely due to their frequent co-occurrence with anomaly

labels during training. This represents a more severe form of shortcut learning, where the model bypasses contextual reasoning and directly maps object presence to anomaly prediction. Our work therefore addresses a VAD-specific challenge: not whether an object exists, but what its presence signifies in a dynamic, context-dependent scene.

### A.6   More Qualitative Examples

To further demonstrate the effectiveness of VAD-DPO in mitigating hallucination and enhancing scene understanding, we provide additional qualitative examples across diverse scenes. By comparing VAD-DPO with baseline models, we highlight its ability to produce context-aware and semantically aligned predictions, especially in cases where conventional LVLMs tend to rely on shortcut correlations.

*System Prompt*: Determine whether an abnormal event occurs in the given video.

*Scene Semantic*: An abnormal event: pedestrians are crossing the road without a crosswalk.

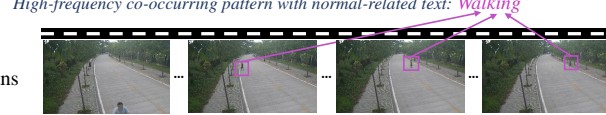

*High-frequency co-occurring pattern with normal-related text: Walking*

*Prediction of Qwen2.5-VL-7B*:
No, a pedestrians is simply walking across the road, and everything appears normal in the scene.

*Prediction of LLaVA-NeXT-Video-7B*:
No, people are casually walking, and there is no indication of any abnormal activity.

*Prediction of LLaVA-1.5-13B*:
No, a walking pedestrian is visible, and nothing unusual seems to be occurring.

*Prediction of InternVL3-8B*:
No, the scene just shows an individual crossing the street on foot, which seems typical.

*Prediction of Ours*:
Yes, a pedestrian is crossing the street without a crosswalk, which constitutes an abnormal event.

*System Prompt*: Determine whether an abnormal event occurs in the given video.

*Scene Semantic*: A normal campus road where a police car is on routine patrol.

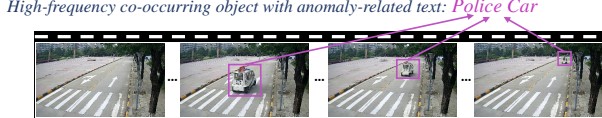

*High-frequency co-occurring object with anomaly-related text: Police Car*

*Prediction of Qwen2.5-VL-7B*:
Yes, a police car is approaching in the scene, suggesting that something abnormal might be taking place.

*Prediction of LLaVA-NeXT-Video-7B*:
Yes, the presence of a police car on the road implies that an unusual incident could be happening..

*Prediction of LLaVA-1.5-13B*:
Yes, the police car's appearance in the scene points toward possible emergency activity.

*Prediction of InternVL3-8B*:
Yes, a police car is approaching in the scene, suggesting that something abnormal might be taking place.

*Prediction of Ours*:
No, the scene shows pedestrians walking calmly in the distance and a police car entering routinely—no abnormal event is occurring.

Figure A.3: Qualitative comparison illustrating hallucination in LVLM-based VAD. While Qwen2.5-VL-7B, LLaVA-NeXT-Video-7B, LLaVA-1.5-13B, and InternVL-8B incorrectly predict an anomaly due to the presence of high-frequency co-occurring objects or patterns, our method accurately identifies the scene as normal by leveraging contextual understanding. Green boxes indicate correct predictions, red boxes indicate incorrect predictions, and purple highlights mark high-frequency co-occurring objects or patterns that may induce hallucination.

## B   Limitations

In this section, we discuss the current limitations of our proposed approach, potential avenues for future improvements, and the broader societal impacts of our work. While our method has

demonstrated significant gains in both anomaly detection and reasoning quality, there remain areas where further research is warranted.

**Limitation 1: Model scale and resource constraints.** Due to computational constraints, our experiments with VAD-DPO are limited to 2B-scale vision-language models. Although results already show strong improvements over larger baselines, it is unclear whether our findings generalize consistently to larger LVLMs or highly parameterized architectures.

**Future Work.** We plan to extend VAD-DPO to larger models such as Qwen2.5-VL-7B or LLaVA-Next-Video-7B, and examine whether preference optimization at scale can yield even more robust mitigation of hallucinations and further enhance reasoning capacity under complex scene semantics.

**Limitation 2: Residual hallucination effects.** Despite effectively reducing shortcut-driven hallucinations, VAD-DPO does not eliminate all forms of hallucination. Beyond co-occurrence bias, hallucinations can still arise from other sources such as strong language priors or modality dominance, where the model overly trusts textual cues even when they conflict with visual evidence.

**Future Work.** Future efforts could investigate hallucination attribution across different causal factors by disentangling the contributions of visual, textual, and cross-modal signals. One promising direction is to incorporate introspective decoding or modality-aware routing mechanisms that explicitly modulate information flow based on semantic consistency across modalities.

**Limitation 3: Shortcut-driven bias and fairness risks.** While VAD-DPO effectively mitigates shortcut reliance by optimizing preference alignment, it does not explicitly address group-sensitive fairness concerns. Shortcut-driven hallucinations—such as associating high-frequency visual cues (e.g., fire, police cars) with anomaly semantics—can lead to biased predictions in surveillance scenarios. Such bias may disproportionately affect certain environments or communities by incorrectly flagging contextually normal activities as anomalous, raising potential fairness and ethical risks in real-world deployment.

**Future Work.** Future research should extend VAD-DPO toward fairness-aware anomaly detection. Promising directions include incorporating demographic-sensitive fairness objectives when annotations include individual-level attributes (e.g., age, clothing, or ethnicity), analyzing disparate false alarm rates across population groups or geographic regions, and integrating preference optimization with fairness-oriented training strategies such as adversarial debiasing or group reweighting. Furthermore, we plan to explicitly connect shortcut-driven hallucination risks with broader societal implications to ensure that robustness and fairness are jointly promoted in safety-critical applications.

## C  Broader Societal Impacts

Our work contributes to improving the robustness and interpretability of video anomaly detection systems, especially in safety-critical applications such as surveillance, autonomous monitoring, and emergency response. By reducing hallucination-induced false alarms and promoting visually grounded reasoning, VAD-DPO can enhance the trustworthiness and reliability of AI-assisted decision-making in real-world deployments. More broadly, this work takes a step toward addressing a core challenge in the deployment of large vision-language models: the misalignment between statistical correlations and semantic correctness. By explicitly training models to prefer semantically consistent responses over shortcut-driven predictions, VAD-DPO offers a principled way to mitigate one of the most pervasive limitations of current multimodal AI systems. In doing so, our approach supports the development of more transparent and accountable AI, where the decision-making process is easier to interpret, verify, and trust. We believe these directions are particularly valuable as foundation models continue to be adopted in domains where model outputs carry real consequences for public safety, resource allocation, and ethical oversight.

Nevertheless, we acknowledge that improved reasoning capabilities in surveillance systems may also raise concerns regarding increased surveillance reach or misuse in privacy-sensitive contexts. To mitigate such risks, all probe data used in this study have been anonymized, with visible faces masked to protect personal identity. We also recognize that masking alone may not be sufficient to eliminate

potential privacy risks; thus, future releases of our probe sets will adopt stronger anonymization techniques such as full-face blurring or obfuscation.

Finally, we emphasize the importance of clearly defining deployment boundaries and ensuring appropriate regulatory oversight. Responsible use of video anomaly detection technologies must prevent potential misuse or overreach in surveillance scenarios that could raise concerns about civil liberties and human rights. Careful consideration of ethical guidelines and governance frameworks will be essential to align technical progress with societal values.

