# OpenReview forum: "Do LVLMs Truly Understand Video Anomalies? Revealing Hallucination via Co-Occurrence Patterns"
_NeurIPS.cc/2025/Conference — NeurIPS 2025 poster_

### Official Review · Reviewer_6ZUx · 2025-06-23

**Clarity:** 2
**Significance:** 3
**Originality:** 2
**Rating:** 4
**Confidence:** 4

**Summary:**

The paper investigates whether Large Vision-Language Models (LVLMs) genuinely understand contextual semantics in Video Anomaly Detection (VAD) or rely on superficial visual-textual co-occurrence patterns from pretraining. Through co-occurrence analysis and controlled probes, the authors reveal a "hallucination" phenomenon: LVLMs frequently misclassify anomalies based on statistical shortcuts (e.g., associating "fire" with anomalies even in benign contexts like barbecues). To mitigate this, they propose VAD-DPO, a direct preference optimization method that trains models on contrastive video pairs with identical objects/scenes but divergent anomaly labels. Experiments on six VAD benchmarks show state-of-the-art results.

**Questions:**

See the detailed comments in weaknesses.

**Ethical Concerns:**

["Major Concern: Human rights (including surveillance)"]

**Final Justification:**

My concerns have been addressed in the rebuttal. I would like to raise my rating.

**Limitations:**

Yes.

**Paper Formatting Concerns:**

None.

**Quality:**

2

**Strengths And Weaknesses:**

Strengths:

- Quality: VAD-DPO outperforms 7B/13B models with only 2B parameters (Table 4). Ablations (Table 6) validate the DPO framework.

- Clarity: Well-structured, with clear figures illustrating key concepts (e.g., Fig 1, 4).

- Significance: This paper tries to address the problem of hallucination in VLMs on the task of video anomaly detection, which has broad implications for multimodal trustworthiness.

- Originality: This paper explores the hallucination problem in the video anomaly detection task.



Weaknesses:

- Quality: (1) I believe that in general scenarios, even some words related to anomalies have meanings that are more related to non-anomalies. Compared to the way statistically analyzing bias in this paper, a more reasonable approach should be to calculate the probability that the semantics that should be non-anomalous are classified as anomalous when there are visual concepts related to anomalous words, and vice versa. It would be even better to use real data rather than constructed data. This way, it can better reflect the real situation. (2) For the probe setting, only 200 samples were generated for analysis. It is insufficient and not convincing.
- Clarity: (1) More details about the analysis of hallucination need to be provided. For example, how do you get the associated image in co-occurrence statistics? And the reasonableness needs to be verified. (2) How are "visually similar but semantically contrasting" video pairs sourced/generated? Provide examples beyond object insertion (e.g., context swaps).
- Originality: (1) Some related works [a][b][c] about hallucination in VLMs have been studied. The VAD task may not be involved, but the differences should be clarified. Besides, it is necessary to compare them to validate the effectiveness of the proposed method. (2) To validate the effectiveness of DPO, other mitigating hallucination-related methods need to be compared.


[a] Multi-Object Hallucination in Vision Language Models
[b] Multi-Modal Hallucination Control by Visual Information Grounding
[c] Mitigating Hallucinations in Large Vision-Language Models with Instruction Contrastive Decoding

---

> ### Author Rebuttal · Authors · 2025-07-31
>
> We sincerely thank Reviewer 6ZUx for the constructive review and encouraging feedback. We especially appreciate your recognition of our **strong performance**, **clear presentation**, **hallucination significance**, and **originality in the VAD setting**. Below, we address the main concerns raised in your review.
>
> ***W1: Quality: (1) Recommends evaluating semantic misclassification probabilities under anomaly-related visual concepts using real data, rather than relying solely on statistical analysis and constructed probes.***
>
> **A1:** We thank the reviewer for this thoughtful and constructive suggestion.
> 1. We agree with the core idea that quantifying semantic misclassification probabilities in the presence of anomaly-related visual concepts is a more direct way to measure co-occurrence bias. In fact, this is exactly what our two-stage analysis aims to achieve:
>
> - **Stage 1 – Hypothesis Formation via Co-occurrence Statistics:** As shown in Table 1, we begin by analyzing large-scale pretraining data to identify visual concepts (e.g., *fire*, *gun*) that frequently co-occur with anomaly-related text. This serves as the hypothesis generation phase, helping us pinpoint likely shortcut triggers.
>
> - **Stage 2 – Hypothesis Testing via Probe Evaluation:** The core of our analysis lies in validating this hypothesis through carefully designed probe datasets (Table 2). This directly aligns with the approach suggested by the reviewer. Specifically, our probe sets are constructed to evaluate conditional misclassification rates—e.g.,  **FPR = P(model predicts "anomaly" | semantics are normal $∧$ shortcut object is present)**. The resulting false positive rates of 70–80% provide direct empirical evidence of co-occurrence-induced hallucination.
>
> 2. We also appreciate the reviewer’s emphasis on evaluating models in real-world settings. Indeed, this is why we evaluate our method on six real-world VAD benchmarks. At the same time, we emphasize that using **constructed data is necessary** for controlled diagnostic evaluation. In real anomaly videos, shortcut objects (e.g., guns) are often entangled with anomalous context (e.g., robbery), making it difficult to isolate causality. Our probe sets serve as **counterfactual test sets**, where inserting a shortcut object into an otherwise normal scene allows us to disentangle causal effects and clearly demonstrate that the model’s false alarm is triggered by the object.
>
> We will clarify this two-stage methodology and its rationale more explicitly in the final version of the paper.
>
>
> ***W2: Quality: (2) For the probe setting, only 200 samples were generated for analysis. It is insufficient and not convincing.***
>
> **A2:** We thank the reviewer for raising this important concern about the sample size of our probe evaluation. We would like to clarify that constructing high-quality video probes requires significant manual effort for editing, screening, and validation.
>
> While we agree that a larger dataset could offer even more robust analysis, we emphasize that our current probe set of 200 samples is already sufficient to reveal an **extremely strong and consistent effect** across all tested models. Crucially, our finding is not merely that "some models make errors," but that **false positive rates often exceed 70%** for anomaly-related objects inserted into normal scenes, as shown in **Table I**. We also include results for domain-specific VAD models such as Holmes-VAU [1] and Hawk [2]. These models exhibit similarly high hallucination rates under the same conditions.
>
> This consistent and severe failure pattern is observed across architectures and model sizes, demonstrating the **systemic nature** of the co-occurrence-induced hallucination problem. Therefore, we believe the current sample size is sufficient and effective for diagnosing and validating the phenomenon. We will clarify this point in the final version of the paper.
>
> ``Table I. Quantitative comparison on hallucination Probe Sets.``
> |Model|Param.|Fire (FPR↓)|Gun (FPR↓)|Blood (FPR↓)|Knife (FPR↓)|Avg. (FPR↓)|Set 2 (FNR↓)|
> |:-:|:-:|:-:|:-:|:-:|:-:|:-:|:-:|
> |Qwen2.5-VL|7B|80.0|65.0|67.5|57.5|67.5|77.5|
> |Qwen2.5-VL|32B|67.5|47.5|60.0|50.0|56.3|60.0|
> |LLaVA-1.5|7B|85.0|80.0|77.5|72.5|78.8|82.5|
> |LLaVA-1.5|13B|75.0|62.5|65.0|60.0|65.6|75.0|
> |LLaVA-NeXT-Video|7B|82.5|70.0|67.5|55.0|68.8|72.5|
> |LLaVA-NeXT-Video|34B|67.5|52.5|62.5|47.5|57.5|57.5|
> |InternVL3|8B|75.0|72.5|65.0|60.0| 68.1|80.0|
> |LAVAD|13B|85.0|82.5|80.0|87.5|82.5|82.5|
> |MMEval|7B|77.5|80.0|85.0|85.0|81.9|80.0|
> |AnomalyRuler|7B|82.5|87.5|90.0|80.0|81.9|82.5|
> |Hawk|7B|90.0|80.0|77.5|72.5|85.0|77.5|
> |Holmes-VAU|2B|75.0|80.0|72.5|72.5| 75.0|80.0|
> |**VAD-DPO (Ours)**|**2B**|**15.0**|**12.5**|**17.5**|**7.5**| **13.1**| **12.5**|
>
>
> [1] Holmes-vau: Towards long-term video anomaly understanding at any granularity. CVPR 2025.\
> [2] Hawk: Learning to understand open-world video anomalies. NeurIPS 2024.
>
>
> ***W3: Clarity: (1) More details about the analysis of hallucination need to be provided. How do you get the associated image in co-occurrence statistics?***
>
> **A3:** We thank the reviewer for pointing out the need for more clarity in our co-occurrence analysis. As described in Section 3.1 (Lines 126–128), we conduct this analysis using the LAION-CC-SBU dataset. We begin by using GPT to generate approximately 3k candidate textual phrases that are potentially associated with anomaly or normality. These phrases are manually reviewed to ensure semantic clarity and proper categorization. We then match these phrases against LAION captions to identify text-image pairs where a caption contains a candidate phrase. For each matched image, we apply YOLOv5 to detect objects and filter out low-confidence detections to ensure statistical robustness.
>
> This process allows us to compute co-occurrence statistics by measuring how frequently certain visual objects appear in images whose captions express either anomaly-related or normality-related concepts. In this way, the co-occurrence patterns are estimated from large-scale real-world data with careful semantic control. We will make this intermediate matching step more explicit in the final version.
>
>
> ***W4: Clarity: (2) How are "visually similar but semantically contrasting" video pairs sourced/generated? Provide examples beyond object insertion.***
>
> **A4:** We apologize for the lack of clarity in the main paper regarding the construction of video pairs. This process is detailed in Appendix (Lines 543–553). In summary, we construct such counter-example preference pairs using two strategies:
>
> - **Intra-scene contrast**: Two clips are drawn from the same environment but differ in semantics due to subtle changes. for example, *walking normally* vs. *sudden erratic movement in a park*.
> - **Cross-scene context shift**: Two clips share the same visual activity (e.g., *driving*), but occur in different contexts that alter their semantic interpretation, e.g., *driving on a pedestrian street (abnormal)* vs. *driving on a road (normal)*.
>
> These pairs are used as training supervision in our VAD-DPO framework to teach the model to prefer semantically correct outputs. Due to rebuttal policy restrictions, we are unable to include examples. We will add a concise visual summary with additional examples in the final version.
>
>
> ***W5: Originality: Clarify differences from related hallucination works and compare with other mitigation methods.***
>
> **A5:** We thank the reviewer for pointing out these relevant works.
>
> 1. **On Related Works:** We thank the reviewer for this sharp observation. We agree that co-occurrence has been studied as a cause of object hallucination in prior VLM works. However, we argue that the nature of the co-occurrence and the resulting failure mode in VAD are fundamentally different, which forms the core focus of our work. Object hallucination typically involves **object–context co-occurrence**. In contrast, VAD hallucination stems from **object–label co-occurrence**. The model does not invent new objects, it correctly detects real objects (e.g., a fire in the scene). However, it incorrectly interprets these objects as anomalous purely based on their frequent co-occurrence with anomaly labels during training. This reflects a more severe form of shortcut learning: the model bypasses contextual reasoning and directly maps object presence to anomaly prediction. Therefore, our work addresses a **VAD-specific challenge**: not whether an object exists, but what its presence means in a dynamic, context-dependent scene.
>
> 2. **On Comparison with Mitigation Methods:** Our ablation study (Table 6) compares VAD-DPO with other training paradigms (e.g., instruction tuning, PPO). Reviewer-cited approaches like contrastive decoding or M3ID are **inference-time decoding strategies** rather than **training-time alignment methods**. Thus, a direct comparison is non-trivial. Nonetheless, to address this concern, we provide results in **Table II** where such decoding-based mitigation methods are compared against our VAD-DPO with Qwen2-VL-2B as backbone. As expected, our training-based method consistently outperforms decoding-time mitigation techniques. We will further clarify the scope of our contribution in the final version.
>
> ``Table II. Comparison of Hallucination Mitigation Methods.``
> |Method|NWPU (AUC↑)|Set 1 (FPR↓)|Set 2 (FNR↓)|
> |:-:|:-:|:-:|:-:|
> |M3ID [3]|69.3|62.5|57.5|
> |ICD [4]|68.6|66.3|62.5|
> |VCD [5]|71.1|53.8|60.0|
> |OPERA [6]|70.2|55.0|62.5|
> |**VAD-DPO (Ours)**|**79.1**| **13.1**| **12.5**|
>
> [3] Multi-modal hallucination control by visual information grounding. CVPR 2024.\
> [4] Mitigating hallucinations in large vision-language models with instruction contrastive decoding. ACL 2024.\
> [5] Mitigating object hallucinations in large vision-language models through visual contrastive decoding. CVPR 2024.\
> [6] Opera: Alleviating hallucination in multi-modal large language models via over-trust penalty and retrospection-allocation. CVPR 2024.

---

> > ### Comment · Reviewer_6ZUx · 2025-08-04
> > **Replying to Rebuttal by Authors**
> >
> > Thanks for the authors' responses. Most of my concerns have been addressed. Regarding W1, I still believe that constructed data is not the best way to explore the problem of hallucination. The constructed data is not always entirely accurate and can lead to some bias. Besides, for the categories of fire, gun, blood, and knife, it is easy to get normal videos from the Internet, taking into account that you don't have a large amount of test data.

---

> ### Author Response · Authors · 2025-08-04
>
> Thank you for your follow-up and for acknowledging that most concerns have been addressed.
>
> Regarding your remaining point on the use of constructed data in W1, we fully understand and appreciate your concern. Indeed, constructed samples may introduce artifacts or biases if not carefully controlled. To mitigate this, we adopted a strict validation process:
>
> - **Manual Screening**: All videos are reviewed by multiple annotators to ensure that the inserted object does not introduce unintended anomaly cues or observable artifacts beyond its intended visual presence.
> - **Exclusion of Ambiguous Cases**: We filter out videos in which the anomaly-related object could reasonably support multiple semantic interpretations (e.g., police cars in tense crowd scenes).
>
> We also agree that real-world videos containing fire, guns, blood, or knives in semantically normal contexts do exist online. However, our goal is not merely to collect such examples, but to **diagnose whether the presence of these high-frequency objects alone triggers hallucinations**. To achieve this, we require **paired samples**: the same video with and without the object, while keeping all other factors constant. In real-world datasets, it is extremely difficult to find hundreds of such tightly controlled video pairs where the only semantic change lies in the presence or absence of one object.
>
> In contrast, our constructed probe sets allow us to systematically isolate this causal factor in a controlled manner. To further minimize bias, we build these probes using scenes from standard VAD benchmarks where the models originally make correct predictions before object insertion.
>
> We will clarify these design choices and validation procedures more explicitly in the final version. In future work, we also plan to explore more realistic generation strategies or collect sufficient real videos to construct semantically controlled probe sets without synthetic modification.
>
> Thank you again for your thoughtful and constructive feedback. If you have any remaining concerns or suggestions, we would be happy to further discuss and clarify them.

---

> ### Author Response · Authors · 2025-08-05
>
> Dear Reviewer 6ZUx,
>
> We are sincerely grateful for your thoughtful and encouraging feedback. In particular, we deeply appreciate your recognition of our strong performance, clear presentation, the significance of hallucination analysis, and the originality of our work in the VAD setting.
>
> Your comments have greatly contributed to improving the clarity and rigor of our paper. We are committed to revising the manuscript accordingly to fully incorporate your suggestions and ensure the highest quality possible.
>
> If there are any remaining concerns or aspects you would like us to further clarify, we would be more than happy to address them. Otherwise, we sincerely hope that the revisions made in response to your feedback may support a favorable final assessment.
>
> Thank you once again for your time, generosity, and kind consideration.
>
> Best regards,\
> Authors of Submission933

---

> > ### Comment · Reviewer_6ZUx · 2025-08-05
> >
> > Thanks for your responses. I have no other concerns at the moment.

---

### Official Review · Reviewer_TxqF · 2025-06-26

**Clarity:** 2
**Significance:** 3
**Originality:** 2
**Rating:** 5
**Confidence:** 4

**Summary:**

This paper investigates whether Large Vision-Language Models (LVLMs) for video anomaly detection (VAD) detect video anomalies based on real understanding of the scene or superficial statistical shortcuts learned during pretraining. To probe this, the authors create two synthetic datasets: one with normal scenes augmented with anomalous objects (e.g., a knife), which appear frequently in the pre-training dataset with anomaly-related phrases, and one with abnormal scenes augmented with normal objects. Their findings reveal that LVLMs often rely on such shortcuts. To address this, they propose VAD-DPO, a direct preference optimization method that encourages LVLMs to focus on the semantics of the scene by promoting better alignment between correct visual-text pairs over semantically different but visually similar pairs with the opposite label. VAD-DPO outperforms both training-free approaches and off-the-shelf LVLMs on six VAD benchmarks.

**Questions:**

1. How does VAD-DPO perform on the probing datasets compared to the off-the-shelf baseline? Additionally, how do LVLMs perform on the probing videos when the anomaly- or normality-related objects are removed?
2. How can you make sure that the generated videos are normal after generating them with anomaly-related objects?
3. What do the GPT-generated textual phrases look like, and how are frame-level anomaly scores computed?
4. What is the novelty of the proposed VAD-DPO method compared to existing approaches that use visually similar but semantically contrasting data?
5. Why did you report AUC-ROC for the XD-Violence dataset instead of the standard AP metric?

**Ethical Concerns:**

["NO or VERY MINOR ethics concerns only"]

**Final Justification:**

The authors addressed my concerns regarding the evaluation of VAD-DPO on the probing datasets and on the same videos without the augmented anomaly/normality-related objects. They clarified that the synthetic videos in these probing sets undergo manual verification, explained how the GPT-generated textual phrases are created (with examples), and described how the anomaly scores are computed. While not entirely new, I also appreciate the proposed strategy to help the model focus on anomalies by using visually similar but semantically different videos. The clarifications and additional results on XD-Violence using the AP metric were helpful. Given these clarifications, I am increasing my score to Accept.

**Limitations:**

yes

**Paper Formatting Concerns:**

No formatting issues

**Quality:**

3

**Strengths And Weaknesses:**

Strengths:
1. The paper addresses a relevant question in video anomaly detection (VAD): whether LVLMs rely on true scene understanding or visual/textual shortcuts associated with normality or abnormality. This is studied by probing LVLMs using synthetic videos augmented with such cues.
2. The authors propose a solution, VAD-DPO, to make the model focus on the scene instead of relying on such shortcuts. The method outperforms larger off-the-shelf LVLMs, demonstrating its effectiveness.
3. The paper is generally well-written and clearly motivates the problem, with valuable insights for the community that should encourage future methods to consider such shortcut learning. The authors claim that the study of these limitations in VAD is a novel contribution.

Weaknesses:
1. Despite its strengths, the paper lacks some evaluations and clarifications. In particular, it does not report how VAD-DPO performs on the probing datasets compared to the off-the-shelf baseline, which would help validate its effectiveness against shortcuts. Also, to better understand the reliance on shortcuts, the same analysis presented in Table 2 could have been repeated on the same synthetic videos without the anomaly/normality-related objects.
2. Some synthetic videos used in the probing datasets may be ambiguous. For example, in Fig. 1, one could expect a fire if there were charcoal under the grill or a pan was used instead of a grill, however, a fire of that size on a grill may seem anomalous.
3. Some implementation details are missing, such as examples of the 3k GPT-generated textual phrases and how frame-level anomaly scores are computed, which limits reproducibility.
4. While the shortcut analysis may be novel for VAD, the proposed training objective, which involves training on visually similar but semantically different videos, is similar to training objectives used for other tasks.
5. Another aspect is that the standard metric on XD-Violence is the area under the precision-recall curve, while the evaluation uses AUC.

---

> ### Author Rebuttal · Authors · 2025-07-31
>
> We sincerely thank Reviewer TxqF for the detailed review and encouraging comments. We particularly appreciate your recognition of our relevant **problem formulation**, **effective DPO solution**, and **clear motivation with community insight**. Below, we address the main concerns raised in your review.
>
> ***W1&Q1: How does VAD-DPO perform on the probing datasets compared to the off-the-shelf baseline? Additionally, how do LVLMs perform on the probing videos when the anomaly- or normality-related objects are removed?***
>
> **A1**: We thank Reviewer TxqF for the insightful question. We address the two aspects below:
>
> 1. **VAD-DPO vs. Off-the-Shelf Baselines on Probing Datasets**: We conduct a dedicated evaluation using our co-occurrence-sensitive probe sets. As described in Appendix A.4 (Table A.1), these probes evaluate whether the model misclassifies scenes due to the presence of high-frequency anomaly-associated visual patterns in misleading contexts. To provide a more comprehensive comparison, we further extend this table to include domain-specific expert models (Holmes-VAU [1], Hawk [2]) and LVLM-based training-free VAD methods (LAVAD [3], MMEval [4], and AnomalyRuler [5]). The updated version is presented below as **Table I**. While even expert models exhibit high error rates (often >75% FPR), our **VAD-DPO slashes the average False Positive Rate to just 13.1%**, a remarkable reduction of over 80% compared to these anomaly-focused models. These findings offer strong quantitative evidence that VAD-DPO successfully mitigates shortcut learning from co-occurrence patterns. We will incorporate this expanded version into the final paper to directly address this concern.
>
> ``Table I. Quantitative comparison on hallucination Probe Sets.``
> |Model|Param.|Fire (FPR↓)|Gun (FPR↓)|Blood (FPR↓)|Knife (FPR↓)|Avg. (FPR↓)|Set 2 (FNR↓)|
> |:-:|:-:|:-:|:-:|:-:|:-:|:-:|:-:|
> |Qwen2.5-VL|7B|80.0|65.0|67.5|57.5|67.5|77.5|
> |Qwen2.5-VL|32B|67.5|47.5|60.0|50.0|56.3|60.0|
> |LLaVA-1.5|7B|85.0|80.0|77.5|72.5|78.8|82.5|
> |LLaVA-1.5|13B|75.0|62.5|65.0|60.0|65.6|75.0|
> |LLaVA-NeXT-Video|7B|82.5|70.0|67.5|55.0|68.8|72.5|
> |LLaVA-NeXT-Video|34B|67.5|52.5|62.5|47.5|57.5|57.5|
> |InternVL3|8B|75.0|72.5|65.0|60.0| 68.1|80.0|
> |LAVAD|13B|85.0|82.5|80.0|87.5|82.5|82.5|
> |MMEval|7B|77.5|80.0|85.0|85.0|81.9|80.0|
> |AnomalyRuler|7B|82.5|87.5|90.0|80.0|81.9|82.5|
> |Hawk|7B|90.0|80.0|77.5|72.5|85.0|77.5|
> |Holmes-VAU|2B|75.0|80.0|72.5|72.5| 75.0|80.0|
> |**VAD-DPO (Ours)**|**2B**|**15.0**|**12.5**|**17.5**|**7.5**| **13.1**| **12.5**|
>
> 1. **Performance Impact of Removing Shortcut Objects**: This is an excellent point. The base videos in Probe Set 1 are manually curated normal scenes selected from standard VAD datasets. Prior to inserting objects, most models tend to assign low anomaly scores. In response to your suggestion, we further remove the inserted objects from the perturbed clips and re-evaluate all models. As shown **in Table II** below, the removal of these misleading objects significantly reduces the false positive rate across models, effectively restoring correct predictions. These results confirm that the hallucinations are indeed triggered by co-occurrence-driven patterns.
>
> ``Table II. FPR (%) on Probe Set 1 after removing misleading objects.``
> |Model|Param.|Fire|Gun|Blood|Knife|Avg.|
> |:-:|:-:|:-:|:-:|:-:|:-:|:-:|
> |Qwen2.5-VL|7B|7.5|5.0|7.5|2.5|5.6|
> |Qwen2.5-VL|32B|2.5|5.0|5.0|0.0|3.1|
> |LLaVA-1.5|7B|15.0|17.5|17.5|12.5|16.3|
> |LLaVA-1.5|13B|12.5|15.0|17.5|7.5|13.1|
> |LLaVA-NeXT-Video|7B|7.5|5.0|10.0|12.5|8.8|
> |LLaVA-NeXT-Video|34B|0.0|7.5|5.0|2.5|3.8|
> |InternVL3|8B|10.0|7.5|12.5|7.5|9.4|
> |LAVAD|13B|12.5|10.0|7.5|10.0|10.0|
> |MMEval|7B|12.5|12.5|7.5|10.0|10.6|
> |AnomalyRuler|7B|12.5|7.5|7.5|10.0|9.4|
> |Hawk|7B|7.5|7.5|2.5|2.5|5.0|
> |Holmes-VAU|2B|5.0|5.0|7.5|5.0|5.6|
> |**VAD-DPO (Ours)**|**2B**|**0.0**|**2.5**|**2.5**|**0.0**| **1.3**|
>
> [1] Holmes-vau: Towards long-term video anomaly understanding at any granularity. CVPR 2025.\
> [2] Hawk: Learning to understand open-world video anomalies. NeurIPS 2024.\
> [3] Harnessing large language models for training-free video anomaly detection. CVPR 2024.\
> [4] Uncovering what why and how: A comprehensive benchmark for causation understanding of video anomaly. CVPR 2024.\
> [5] Follow the rules: Reasoning for video anomaly detection with large language models. ECCV 2024.
>
> ***W2&Q2: How can you make sure that the generated videos are normal after generating them with anomaly-related objects?***
>
> **A2**: We thank the reviewer for this thoughtful question. Our goal in constructing probe samples is to simulate scenes where the most plausible and commonsense interpretation is normal, despite the presence of anomaly-related objects. For example, a video depicting a man barbecuing in a backyard may include a "fire" object, yet the overall scene is semantically normal. To ensure the normality, we applied a strict human validation protocol:
>
> - **Manual Screening:** All videos are reviewed by multiple annotators to ensure that the inserted object does not introduce unintended anomaly cues beyond its visual presence.
> - **Exclusion of Ambiguous Cases:** We filter out videos in which the anomaly-related object could reasonably support multiple semantic interpretations (e.g., police cars in tense crowd scenes).
>
> While we acknowledge a degree of subjectivity is inevitable, this validation protocol is designed to minimize such ambiguity. We will add a detailed description to the final paper.
>
>
>
> ***W3&Q3: What do the GPT-generated textual phrases look like, and how are frame-level anomaly scores computed?***
>
> **A3:** We thank the reviewer for this excellent question. Below we address the two components:
>
> 1. **GPT-Generated Phrases**: To construct our probing datasets, we use GPT-4 to generate a diverse set of 3k candidate textual phrases related to anomaly and normality. We prompt the model with instructions such as: `Generate a list of concise phrases that describe either abnormal or normal events in video scenes.`  Some example outputs include:
> - Anomaly-related: *"a person holding a knife"*, *"fire spreading rapidly"*, *"man lying motionless on the ground"*
> - Normality-related: *"people walking across a street"*, *"children playing in a park"*, *"car driving along a road"*
>
> We manually review and filter these phrases to ensure semantic clarity and diversity.
>
> 2. **Anomaly Score Computation**: Our evaluation protocol follows LAVAD [3] and consists of two stages:
>
> - **Frame Sampling and Scoring:** For each video segment, we uniformly sample 8 frames and pair each with a scoring-style prompt, consistent with LAVAD [5]. For example, we ask the LVLM: *“If you were a law enforcement agency, how would you rate the scene described on a scale from 0 to 1, with 0 representing a standard scene and 1 denoting a scene with suspicious activities?”* The model then directly outputs a numerical score (e.g., 0.83), which serves as the *initial anomaly score* for that frame.
> - **Score Refinement and Frame-wise Assignment:** Then, we refine these sparse scores through a weighted aggregation mechanism. Specifically, we leverage temporal attention weights from the decoding process to smoothly propagate scores from sampled frames to their neighboring frames. This produces a dense per-frame anomaly score sequence covering the entire video.
>
> We will include this procedure in the final version to enhance reproducibility.
>
>
> ***W4&Q4: What is the novelty of the proposed VAD-DPO method compared to existing approaches that use visually similar but semantically contrasting data?***
>
> **A4:** We appreciate the reviewer’s question regarding the novelty of our method. We acknowledge that the idea of learning from visually similar but semantically contrasting data is not entirely new. However, our contribution lies in applying this idea to a **specific and underexplored problem**, namely, shortcut learning caused by co-occurrence in **VAD**. Our novelty stems from the following aspects:
>
> - **Problem Definition in VAD**: To the best of our knowledge, we are the first to identify and formalize co-occurrence-induced hallucinations as a critical failure mode in LVLM-based VAD.
> - **Systematic Diagnosis of the Problem**: We provide a multi-level analysis of co-occurrence shortcuts (e.g., object, combination, interaction) and design targeted probe sets to quantify the extent of hallucination.
> - **Tailored Solution via VAD-DPO**: We introduce a framework that directly mitigates the problem. The preference-pair design is carefully aligned with the contextual semantics of anomaly detection and the specific failure patterns of LVLMs.
>
> In summary, the **novelty of our work lies in the identification, formalization, and resolution of a overlooked problem in VAD**. We believe this form of contribution is essential for advancing reliable deployment of LVLMs in VAD.
>
>
> ***W5&Q5: Why did you report AUC for the XD-Violence instead of the standard AP metric?***
>
> **A5**: We thank the reviewer for pointing out this important detail. We fully acknowledge that AP is a widely used metric for XD-Violence. Compared to AUC, this metric mainly focuses on the performance of VAD method in identifying anomalous events. In other words, AP pays attention to classifying the anomaly correctly rather than the overall separation. To address this, we conduct additional experiments and report the frame-level AP scores on XD-Violence in **Table III** below. VAD-DPO consistently outperforms all baselines under this metric as well, further validating its effectiveness. Note that the original Holmes-VAU paper reports video-level results; we therefore re-evaluated its frame-level AP using the official released model weights for a fair comparison.
>
> ``Table III. Frame-level AP(%) Comparasion on XD-Violence.``
> |Model|Param.|XD|
> |:-:|:-:|:-:|
> |Qwen2.5-VL|7B|65.24|
> |LAVAD|13B|62.01|
> |MMEval|7B|69.65|
> |AnomalyRuler|7B|68.22|
> |Hawk|7B|75.39|
> |Holmes-VAU|2B|83.47|
> |**VAD-DPO (Ours)**|**2B**|**84.53**|

---

> > ### Comment · Reviewer_TxqF · 2025-08-04
> >
> > I appreciate the detailed response from the authors. I have read both the authors’ replies and the other reviewers’ comments.
> >
> > **W1 & Q1**
> > I was specifically referring to the baseline with Qwen2.5-2B (or Qwen2.5-3B?). That said, it seems that models with stronger LLM decoders have less bias, so I understand the decision not to report those results.
> > Thank you for showing that VAD-DPO performs better on the videos in *probe set 1* not augmented with anomalous objects.
> >
> > **W2 & Q2**
> > I appreciate the clarification regarding the manual inspection process to make sure that the probe samples are semantically correct. Do you perform this manual validation for both *probe set 1* and *probe set 2*?
> >
> > **W3 & Q3**
> > Thank you as well for explaining how the anomaly scores are computed. I have a couple of follow-up questions:
> >
> > 1. What threshold do you use across different models to compute the false positive and false negative rates?
> > 2. Is Figure 4 simply an illustrative example of the model output, which is not used in the evaluation?
> >
> > **W4 & Q4**
> > I am fine with the paper’s main contribution being the analysis of how LVLMs may rely on shortcuts rather than reasoning about anomalies in context. I would suggest clarifying in the final manuscript which models are training-free and which are training-based, and also highlighting the differences between the vision-language models used.
> >
> > **W5 & Q5**
> > Thank you for including the AP results on XD-Violence.

---

> ### Author Response · Authors · 2025-08-04
>
> Thank you for the encouraging follow-up and for highlighting important clarifications.
>
> - **Manual Validation of Probe Sets**: We confirm that manual validation is conducted for both Probe Set 1 and Probe Set 2. Each sample is reviewed by multiple annotators to ensure the intended semantic interpretation holds (e.g., the anomaly in Set 2 remains recognizable despite the presence of normal-looking objects). Ambiguous or semantically uncertain cases are filtered out to ensure consistency.
>
> - **Threshold for FPR/FNR**: We use a **fixed threshold of 0.5** across all models to compute false positive and false negative rates, ensuring fair comparison despite differences in output score distributions. We also experimented with adaptive thresholds (e.g., mean + standard deviation), and found that the relative trends remained consistent.
>
> - **Clarification on Figure 4**: Figure 4 serves purely as an **illustrative example** of shortcut-induced hallucinations. It is **not included in any quantitative evaluation**, which is based entirely on the systematically constructed probe sets introduced in Section 3.1 and Appendix A.1.
>
> - **Planned Revisions**: In response to your suggestions, we will explicitly indicate in the final version which models are training-free and which are training-based.
>
> We will incorporate all of these clarifications into the final version. Thank you again for your thoughtful and constructive feedback. If you have any further questions or suggestions, we would be happy to provide additional clarifications.

---

> ### Author Response · Authors · 2025-08-05
>
> Dear Reviewer TxqF,
>
> We sincerely thank you for your thoughtful and detailed follow-up. We truly appreciate your recognition of the core contribution and your insightful suggestions, which have greatly helped us improve the paper.
>
> Following your comments, we have provided additional clarifications regarding manual validation across both probe sets, the thresholding strategy for FPR/FNR, the illustrative role of Figure 4, and our planned revisions to clearly distinguish training-free and training-based models in the final version.
>
> If there are any remaining concerns or aspects that you feel would benefit from further clarification, we would be more than happy to address them. Otherwise, we sincerely hope our responses have addressed your comments satisfactorily.
>
> Thank you once again for your time, thoughtful engagement, and constructive feedback.
>
> Best regards,\
> Authors of Submission933

---

> > ### Comment · Reviewer_TxqF · 2025-08-06
> >
> > Thank you to the authors for addressing all my concerns. I have no further questions.

---

### Official Review · Reviewer_BrW1 · 2025-07-02

**Clarity:** 4
**Significance:** 3
**Originality:** 3
**Rating:** 4
**Confidence:** 4

**Summary:**

This paper focuses on tackling the **hallucination issues** in video anomaly understanding and identifies **co-occurrence patterns between visual elements and textual descriptions** as the root cause of the problem. Specifically, these co-occurrence patterns lead LVLMs to make incorrect predictions when high-frequency visual objects appear in semantically mismatched contexts, as the models rely on statistical shortcuts rather than true scene understanding. To address this challenge, the authors propose a novel **Direct Preference Optimization (DPO)-based approach**. By constructing visually similar but semantically contrasting video pairs, this method encourages the model to make predictions that are aligned with scene semantics instead of co-occurrence patterns. Extensive experiments demonstrate that this DPO-based strategy significantly enhances LVLMs’ capabilities in anomaly detection and reasoning, particularly by improving their scene-semantic grounding and reducing hallucination effects.

**Questions:**

See the weaknesses.

Besides, the authors should clearly explain how the performance of these multimodal large models is calculated for anomaly detection tasks. This clarification would greatly enhance the reproducibility of the research.

**Ethical Concerns:**

["NO or VERY MINOR ethics concerns only"]

**Final Justification:**

I maintain my current score.

**Limitations:**

See the weaknesses.

**Quality:**

3

**Strengths And Weaknesses:**

**Strengths:**
1. **Comprehensive Problem Analysis**: The authors present a thorough and insightful examination of the hallucination problem in anomaly scenarios for LVLMs, making a valuable contribution to the field.
2. **Well-Founded Methodology**: The proposed DPO-based approach (VAD-DPO) is well-motivated, with strong logical reasoning underpinning its design and implementation.

**Weaknesses:**
1. **Limited Data Scale**: The training dataset size (~1K preference pairs) may be insufficient for effective DPO alignment, potentially compromising the model's generalization ability.
2. **Insufficient Model Comparisons**:
   - The evaluation lacks a comparison with **domain-specific LVLMs** (e.g., Hawk [25], Holmes-VAU [39]), which should be included as baselines in Table 2 and Table 5.
   - Comparisons with general-purpose LVLMs (e.g., Qwen-VL) are less convincing than direct comparisons against state-of-the-art anomaly-focused models.
3. **Inadequate Validation of Co-Occurrence Patterns**: The authors fail to sufficiently demonstrate the alleviation of Co-Occurrence Patterns through experimental results. In the current version, only final results are provided, without a detailed analysis of the extent to which the Co-Occurrence was addressed.

---

> ### Author Rebuttal · Authors · 2025-07-31
>
> We sincerely thank Reviewer BrW1 for the thorough review and constructive comments. We particularly appreciate your recognition of our **comprehensive problem analysis** and **well-founded methodology**, which we believe are central to our contributions. Below, we address the main concerns raised in your review.
>
> ***W1: Limited Data Scale.***
>
> **A1**: We thank the reviewer for raising this important point. We acknowledge that a dataset of ~1K preference pairs is relatively small in scale. However, the effectiveness of DPO relies not only on quantity, but also on the **semantic clarity and contrastiveness** of the preference pairs. As described in Section 4.2 and Appendix A.3 of the Supplementary Material, our preference pairs are carefully constructed to maximize semantic contrast while minimizing visual confounds, thus providing strong and unambiguous optimization signals.
>
> Moreover, we have evaluated the resulting VAD-DPO model on **four general-purpose video understanding benchmarks**: MVBench [1], PerceptionTest [2], EgoSchema [3], and Video-MME [4]. These benchmarks span diverse domains, including egocentric actions, schematic reasoning, and multi-modal video QA. As shown in Table I, **VAD-DPO achieves a slight improvement** over the original Qwen2-VL-2B, despite not being trained on any of these datasets. This indicates that preference tuning not only preserves the model's generalization ability, but may also enhance it by improving semantic grounding. These results demonstrate that VAD-DPO maintains robust generalization even when trained on a relatively small yet carefully constructed set of preference pairs.
>
> ``Table I. Overall performance (%) comparison on general-purpose video benchmarks.``
> |Benchmark|Qwen2-VL-2B|VAD-DPO (Ours)|
> |:---:|:---:|:---:|
> |MVBench|63.2|**63.6**|
> |PerceptionTest|53.9|**55.0**|
> |EgoSchema|54.9|**55.1**|
> |Video-MME|60.4|**61.2**|
>
> [1] Mvbench: A comprehensive multi-modal video understanding benchmark. CVPR 2024.\
> [2] Perception test: A diagnostic benchmark for multimodal video models. NeurIPS 2023.\
> [3] Egoschema: A diagnostic benchmark for very long-form video language understanding. NeurIPS 2023.\
> [4] Video-mme: The first-ever comprehensive evaluation benchmark of multi-modal LLMs in video analysis. CVPR 2025.
>
>
> ***W2: Insufficient Model Comparisons.***
>
> **A2:** We thank the reviewer for this valuable suggestion. We acknowledge that our initial version lacked direct comparisons with domain-specific LVLMs. To address this, we have conducted additional experiments under the same settings as Table 2 and Table 5 in the main paper.
>
> **Table II** supplements the results in Table 2 by including both domain-specific expert models (Holmes-VAU [5] and Hawk [6]), which are fine-tuned on VAD datasets, and training-free LVLM-based VAD methods (LAVAD [7], MMeval [8], and AnomalyRuler [9]). All results are obtained using the released code and checkpoints, without any retraining or manual tuning.
>
> Despite being trained specifically for VAD, the expert models exhibit substantial hallucination rates. We attribute this to the limited amount of true anomaly samples in current VAD datasets, which encourages the model to rely on spurious co-occurrence patterns during supervised fine-tuning. In contrast, **VAD-DPO reduces the average false positive rate from over 75% in expert models such as Holmes-VAU and Hawk to just 13.1%**, a remarkable reduction of over 80%, demonstrating its unique ability to move beyond such shortcut reliance. Additionally, the training-free methods suffer from severe hallucinations as well, which is expected given their reliance on earlier-generation LVLMs. For instance, LAVAD uses BLIP2 for vision encoding with LLaMA-2-13B, MMeval is based on Vicuna-7B, and AnomalyRuler employs Mistral-7B-Instruct-v0.2. These models generally fall short compared to our approach due to limited multi-modal reasoning capability.
>
> ``Table II. Quantitative comparison on hallucination probe sets.``
> |Model|Param.|Fire (FPR↓)|Gun (FPR↓)|Blood (FPR↓)|Knife (FPR↓)|Avg. (FPR↓)|Set 2 (FNR↓)|
> |:---:|:---:|:---:|:---:|:---:|:---:|:---:|:---:|
> |Qwen2.5-VL|7B|80.0|65.0|67.5|57.5|67.5|77.5|
> |Qwen2.5-VL|32B|67.5|47.5|60.0|50.0|56.3|60.0|
> |LLaVA-1.5|7B|85.0|80.0|77.5|72.5|78.8|82.5|
> |LLaVA-1.5|13B|75.0|62.5|65.0|60.0|65.6|75.0|
> |LLaVA-NeXT-Video|7B|82.5|70.0|67.5|55.0|68.8|72.5|
> |LLaVA-NeXT-Video|34B|67.5|52.5|62.5|47.5|57.5|57.5|
> |InternVL3|8B|75.0|72.5|65.0|60.0| 68.1|80.0|
> |LAVAD|13B|85.0|82.5|80.0|87.5|82.5|82.5|
> |MMEval|7B|77.5|80.0|85.0|85.0|81.9|80.0|
> |AnomalyRuler|7B|82.5|87.5|90.0|80.0|81.9|82.5|
> |Hawk|7B|90.0|80.0|77.5|72.5|85.0|77.5|
> |Holmes-VAU|2B|75.0|80.0|72.5|72.5| 75.0|80.0|
> |**VAD-DPO (Ours)**|**2B**|**15.0**|**12.5**|**17.5**|**7.5**| **13.1**| **12.5**|
>
> **Table III** further compares VAD-DPO with domain-specific models on the HIVAU-70K benchmark. To ensure a fair comparison, we also fine-tuned Hawk on the training split of HIVAU-70K and report its post-tuning performance alongside Holmes-VAU and our VAD-DPO. Again, our method achieves clear gains, confirming its advantage even with anomaly-focused models. We will include these extended comparisons in the final version of the paper to ensure a more complete evaluation.
>
> ``Table III. Performance on HIVAU-70K across different metrics. (C: Caption, E: Event, V: Video)``
>
> |Method|Params|BLEU-C|BLEU-E|BLEU-V|CIDEr-C|CIDEr-E|CIDEr-V|ROUGE-C|ROUGE-E| ROUGE-V|
> |:---:|:---:|:---:|:---:|:---:|:---:|:---:|:---:|:---:|:---:|:---:|
> |LAVAD|13B|0.236|0.115|0.127|0.138|0.093|0.104|0.109|0.105|0.072|
> |MMEval|7B|0.209|0.130|0.135|0.152|0.146|0.112|0.138|0.092|0.103|
> |AnomalyRuler|7B|0.253|0.088|0.122|0.069|0.086|0.162|0.194|0.117|0.108|
> |Hawk|7B|0.833|0.556|0.347|0.328|0.979|1.064|0.281|0.299|0.284|
> |Holmes-VAU|2B|0.913|0.804|0.566|0.467|1.519|1.437|0.329|0.370|0.355|
> |**VAD-DPO**|**2B**|**0.923**|**0.881**|**0.645**|**0.688**|**1.984**|**1.832**|**0.561**|**0.596**|**0.573**|
>
>
> [5] Holmes-vau: Towards long-term video anomaly understanding at any granularity. CVPR 2025.\
> [6] Hawk: Learning to understand open-world video anomalies. NeurIPS 2024.\
> [7] Harnessing large language models for training-free video anomaly detection. CVPR 2024.\
> [8] Uncovering what why and how: A comprehensive benchmark for causation understanding of video anomaly. CVPR 2024.\
> [9] Follow the rules: Reasoning for video anomaly detection with large language models. ECCV 2024.
>
>
>
> ***W3: Inadequate Validation of Co-Occurrence Patterns.***
>
> **A3**: We sincerely thank the reviewer for this insightful observation. We agree that demonstrating the alleviation of co-occurrence-induced errors is essential to validating the effectiveness of our method.
>
> In fact, we conducted a dedicated experiment using our co-occurrence-sensitive probe sets. As described in Appendix A.4 (Table A.1) of the Supplementary Material, these probes evaluate whether the model misclassifies scenes due to the presence of high-frequency anomaly-associated visual patterns in misleading contexts. Following your valuable suggestion, we have now expanded this analysis to include domain-specific VAD models, including Holmes-VAU and Hawk, as summarized in **Table II in A2**. While even expert models exhibit high error rates (often >75% FPR), our **VAD-DPO slashes the average False Positive Rate to just 13.1%**, a remarkable reduction of over 80% compared to these anomaly-focused models. These findings offer strong quantitative evidence that VAD-DPO successfully mitigates shortcut learning from co-occurrence patterns.
>
> We will include this comprehensive comparison in the final revision of the paper to ensure a transparent evaluation.
>
> ***Q1: The authors should clearly explain how the performance of these multimodal large models is calculated for anomaly detection tasks. This clarification would greatly enhance the reproducibility of the research.***
>
> **A4**: We sincerely thank the reviewer for this crucial question regarding reproducibility. Our evaluation protocol follows the established precedent set by prior work [7], and consists of two main stages for deriving quantitative anomaly scores from LVLMs:
>
> - **Frame Sampling and Scoring:** For each video segment, we uniformly sample 8 frames and pair each with a scoring-style prompt, consistent with LAVAD [5]. For example, we ask the LVLM: *“If you were a law enforcement agency, how would you rate the scene described on a scale from 0 to 1, with 0 representing a standard scene and 1 denoting a scene with suspicious activities?”* The model then directly outputs a numerical score (e.g., 0.83), which serves as the *initial anomaly score* for that frame.
> - **Score Refinement and Frame-wise Assignment:** Following [5], we refine these sparse scores through a **weighted aggregation mechanism**. Specifically, we leverage temporal attention weights from the decoding process to smoothly propagate scores from sampled frames to their neighboring frames. This produces a *dense per-frame anomaly score sequence* covering the entire video.
>
> The resulting per-frame scores are directly compared with ground-truth binary frame-level annotations to compute the AUC using the standard ROC formulation. This approach ensures that our evaluation remains consistent with prior works. We will include this procedure in the final version of the paper to enhance reproducibility.

---

> ### Author Response · Authors · 2025-08-06
>
> Dear Reviewer BrW1,
>
> I hope this message finds you well. We sincerely thank you for your thoughtful and constructive feedback. We have carefully prepared and submitted our rebuttal addressing your insightful comments and suggestions. If there are any remaining questions or additional clarification required, we remain eager and open to further discussion. We respectfully look forward to your continued evaluation and response. Thank you once again for your time and effort.
>
> Best regards,\
> Authors of Submission933

---

### Official Review · Reviewer_b99r · 2025-07-06

**Clarity:** 4
**Significance:** 3
**Originality:** 3
**Rating:** 6
**Confidence:** 5

**Summary:**

This paper reveals a hallucination phenomenon in LVLM-based VAD: LVLM tends to rely on statistical shortcuts learned during pre-training rather than performing reasoning when detecting anomalies. Hence, this paper proposes VAD-DPO to train models on visually similar videos with contrasting anomaly labels, encouraging alignment with semantic consistency rather than co-occurrence shortcuts. Experiments show the effectiveness of the proposed method.

**Questions:**

In Table 6, what is the difference between DPO and VAD-DPO?

**Ethical Concerns:**

["NO or VERY MINOR ethics concerns only"]

**Final Justification:**

I am satisfied with the rebuttal, which has addressed my concerns. I'm raising my rating to 6: Strong Accept. This is a great paper and is recommended for presentation at the venue.

**Limitations:**

See weaknesses

**Quality:**

3

**Strengths And Weaknesses:**

Strengths:
1. This paper reveals the shortcut phenomenon of existing VAD evaluation and conducts an empirical analysis. This is an important finding for the VAD research community.
2. The evaluation is comprehensively conducted on six different datasets.
3. This paper is well-written, well-organized, and easy to follow.

Weaknesses:
1. The connection between the revealed shortcut phenomenon and the proposed VAD-DPO method is not strong; it feels more like two separate contributions. How DPO addresses the shortcut issue should be better demonstrated.
2. The proposed method considers only the naive DPO algorithm. More advanced approaches, such as GRPO [r1], or a newly proposed VAD-specific algorithm, could be explored.
3. The proposed method is only evaluated on the Qwen2-VL backbone, which is insufficient. At least one additional backbone, such as InternVL, should be considered.
4. Figure 4 provides a good qualitative illustration. However, a quantitative comparison is needed to demonstrate VAD-DPO's advantage in hallucination cases.

Justification:

This paper identifies the shortcut phenomenon in existing VAD evaluations, which is a meaningful contribution. However, several concerns remain. I suggest a Borderline Accept at this stage.

[r1] Shao, Zhihong, et al. "Deepseekmath: Pushing the limits of mathematical reasoning in open language models." arXiv preprint arXiv:2402.03300 (2024).

---

> ### Author Rebuttal · Authors · 2025-07-31
>
> We sincerely thank Reviewer b99r for the thorough review and constructive comments. We particularly appreciate your recognition of our **shortcut revelation**, **comprehensive evaluation**, and **clarity of writing**. Below, we address the main concerns raised in your review.
>
> ***W1: Clarify the connection between the shortcut phenomenon and the proposed VAD-DPO method.***
>
> **A1**: Thank you for highlighting this concern. VAD-DPO is designed *specifically* to address the shortcut issue we identified. The core idea is to construct **preference pairs $(y_w, y_l)$ that are visually similar but semantically contrasting**, as discussed in Section 4.2. For example, both "a person hitting another in a street alley" and "a person hitting another in a boxing ring" involve the same high-frequency interaction pattern (hitting), which often co-occurs with anomaly-related phrases. However, only the former is truly anomalous. By training the model to prefer the semantically correct output $y_w$ (i.e., boxing ring is normal) over the incorrect $y_l$ (i.e., street fight is abnormal), VAD-DPO explicitly forces the model to go beyond surface-level correlations and reason about **scene semantics**. This mechanism directly targets and mitigates co-occurrence-driven hallucinations. To further demonstrate this connection, we will revise Section 1 (Introduction) and Section 4 (Method) to highlight the causal link between co-occurrence shortcuts and the VAD-DPO.
>
> Importantly, we already provide **quantitative evidence** of this mitigation effect in **Appendix A.4 of the Supplementary Material**. To make this clearer, we also present an **expanded comparison table (Table III in our response to W4)**, which highlights the mitigation effect across multiple models. Specifically, VAD-DPO reduces false positive rates on Probe Set 1 from 60–80% (in baseline LVLMs) to 13%, and false negative rates on Probe Set 2 from 70–80% to 12.5%. These results directly confirm that VAD-DPO alleviates shortcut reliance. We will incorporate these findings into the main paper for clarity..
>
> ***W2: More advanced approaches, such as GRPO, or a newly proposed VAD-specific algorithm, could be explored.***
>
> **A2**: We thank the reviewer for this insightful suggestion. Indeed, GRPO and other advanced variants of DPO provide valuable directions for future exploration. Our primary goal in this paper is to **systematically identify and address the co-occurrence-induced hallucination phenomenon in LVLM-based VAD**, which, to the best of our knowledge, has not been studied before. To establish a solid and interpretable foundation, we adopt the DPO formulation, which directly optimizes semantic preference and offers a stable framework for counter-example training.
>
> To empirically address your suggestion, we conducted new experiments adapting the GRPO algorithm to our VAD task. The results are summarized in Table I below. While GRPO improves over baseline LVLMs, **it underperforms compared to VAD-DPO**. We attribute this to two main reasons: (1) The core challenge of our task lies in distinguishing between visually similar but semantically contrasting videos, which aligns more naturally with the pairwise formulation of DPO than the group-wise ranking used in GRPO. (2) VAD-DPO integrates an Anchored Loss, which is crucial for stabilizing training and preventing the model from forgetting correct semantics. This VAD-specific design leads to better results, while GRPO lacks such task-aware alignment mechanisms.
>
> These results validate the value of our approach, highlighting the importance of designing solutions specifically tailored to VAD. We plan to further explore VAD-specific alignment strategies that account for contextual semantics in future work.
>
>
> ``Table I. Performance comparison on VAD benchmarks and hallucination probe sets across Baseline, GRPO, and VAD-DPO.``
>
> |LVLM|Params|SHTech (AUC↑)|UCF (AUC↑)|XD (AUC↑)|Campus (AUC↑)|MSAD (AUC↑)|Probe Set 1 (FPR↓) |Probe Set 2 (FNR↓)|
> |:-:|:-:|:-:|:-:|:-:|:-:|:-:|:-:|:-:|
> |Qwen2.5-VL|7B|79.4|78.8|83.2|71.9|75.9|67.5|77.5|
> |LLaVA-1.5|13B|76.3|72.8|79.6|70.3|75.1|65.6|75.0|
> |Qwen2-VL|2B|74.7|71.1|76.8|68.9|74.0|72.5|80.0|
> |GRPO (Qwen2-VL)|2B|79.9|77.4|84.2|75.3|78.6|45.0|32.5|
> |**VAD-DPO (Qwen2-VL)**|**2B**|**87.2**|**86.2**|**88.5**|**79.1**|**85.4**|**13.1**|**12.5**|
>
>
>
> ***W3: The proposed method is only evaluated on the Qwen2-VL backbone, which is insufficient. At least one additional backbone, such as InternVL, should be considered.***
>
> **A3:** We thank the reviewer for the valuable suggestion. To address this concern, we have conducted new experiments by applying our VAD-DPO method to the InternVL3-1B. We selected InternVL3-1B for fine-tuning to maintain a comparable parameter scale to Qwen2-VL-2B, thereby ensuring a fair comparison under similar resource constraints. The results are summarized in Table II below.
>
> Specifically, VAD-DPO improves the frame-level AUC of InternVL3-1B by **over 10%** on average across VAD benchmarks, and reduces hallucination-induced error rates on Probe Set 1 from 75.0% to 24.4%, and on Probe Set 2 from 80.0% to 27.5%. Despite having significantly fewer parameters, the fine-tuned InternVL3-1B with VAD-DPO **outperforms the off-the-shelf InternVL3-8B** on both standard VAD benchmarks and hallucination-sensitive probe sets. These results demonstrate that our method is not only effective on Qwen-based models, but also generalizes well to other LVLM architectures. Overall, VAD-DPO emerges as a generalizable optimization framework capable of improving performanc across different backbones.
>
> ``Table II. Performance of InternVL3-1B before and after VAD-DPO, compared to InternVL3-8B.``
>
> |LVLM|Params|SHTech (AUC↑)|UCF (AUC↑)|XD (AUC↑)|Campus (AUC↑)|MSAD (AUC↑)|Probe Set 1 (FPR↓) |Probe Set 2 (FNR↓)|
> |:-:|:-:|:-:|:-:|:-:|:-:|:-:|:-:|:-:|
> |InternVL3|8B|79.6|77.9|83.0|71.2|75.4|70.0|75.0|
> |InternVL3|1B|74.2|71.6|75.7|69.3|72.2|75.0|80.0|
> |**VAD-DPO (InternVL3)**|**1B**|**85.9**|**85.4**|**87.3**|**76.7**|**82.1**|**24.4**|**27.5**|
>
>
>
> ***W4: A quantitative comparison is needed to demonstrate VAD-DPO's advantage in hallucination cases.***
>
> **A4**: We sincerely thank the reviewer for this constructive suggestion. We fully agree that quantitative evaluation is essential to demonstrate the effectiveness of our method in hallucination scenarios.
>
> In fact, we have already conducted a dedicated analysis and reported the results in Table A.1 of Appendix A.4. Due to space constraints, this important result was initially placed in the supplementary material. In that table, VAD-DPO reduces the hallucination-induced false positive rate on Probe Set 1 from ~70% to **13.1%**, and the false negative rate on Probe Set 2 from ~80% to **12.5%**, clearly demonstrating its ability to mitigate shortcut-driven hallucinations.
>
> We appreciate the reviewer’s suggestion to highlight this result more prominently. Accordingly, we will move this analysis into the main paper in the final version. Furthermore, to provide a more comprehensive comparison, we extend the original table by incorporating both domain-specific expert models (Holmes-VAU [1] and Hawk [2]) and LVLM-based training-free VAD methods, including LAVAD [3], MMEval [4], and AnomalyRuler [5]. The updated version is presented below as **Table III**.
>
> ``Table III. Quantitative comparison on hallucination probe sets.``
> |Model|Param.|Fire (FPR↓)|Gun (FPR↓)|Blood (FPR↓)|Knife (FPR↓)|Avg. (FPR↓)|Set 2 (FNR↓)|
> |:-:|:-:|:-:|:-:|:-:|:-:|:-:|:-:|
> |Qwen2.5-VL|7B|80.0|65.0|67.5|57.5|67.5|77.5|
> |Qwen2.5-VL|32B|67.5|47.5|60.0|50.0|56.3|60.0|
> |LLaVA-1.5|7B|85.0|80.0|77.5|72.5|78.8|82.5|
> |LLaVA-1.5|13B|75.0|62.5|65.0|60.0|65.6|75.0|
> |LLaVA-NeXT-Video|7B|82.5|70.0|67.5|55.0|68.8|72.5|
> |LLaVA-NeXT-Video|34B|67.5|52.5|62.5|47.5|57.5|57.5|
> |InternVL3|8B|75.0|72.5|65.0|60.0| 68.1|80.0|
> |LAVAD|13B|85.0|82.5|80.0|87.5|82.5|82.5|
> |MMEval|7B|77.5|80.0|85.0|85.0|81.9|80.0|
> |AnomalyRuler|7B|82.5|87.5|90.0|80.0|81.9|82.5|
> |Hawk|7B|90.0|80.0|77.5|72.5|85.0|77.5|
> |Holmes-VAU|2B|75.0|80.0|72.5|72.5| 75.0|80.0|
> |**VAD-DPO (Ours)**|**2B**|**15.0**|**12.5**|**17.5**|**7.5**| **13.1**| **12.5**|
>
> [1] Holmes-vau: Towards long-term video anomaly understanding at any granularity. CVPR 2025.\
> [2] Hawk: Learning to understand open-world video anomalies. NeurIPS 2024.\
> [3] Harnessing large language models for training-free video anomaly detection. CVPR 2024.\
> [4] Uncovering what why and how: A comprehensive benchmark for causation understanding of video anomaly. CVPR 2024.\
> [5] Follow the rules: Reasoning for video anomaly detection with large language models. ECCV 2024.
>
> ***Q1:In Table 6, what is the difference between DPO and VAD-DPO?***
>
> **A5**: Thank you for the thoughtful question. The key difference lies in our loss function design, as detailed in Equations (3)–(6) of the paper.
>
> Standard DPO uses the contrastive loss in Equation (3), which encourages the model to assign higher probability to the preferred output (**$y_w$**) over the less preferred one (**$y_l$**). Our proposed VAD-DPO, defined in Equation (6), extends this by adding an additional **anchored loss term $L_{Anc}$** (Equation (5)), which is scaled by a hyperparameter $γ$. The purpose of $L_{Anc}$ is to **explicitly encourage the model to maintain a strong preference for semantically correct samples**, even in the absence of direct contrast. This helps improve training stability, which is especially important in VAD due to the subtle and context-dependent nature of semantic alignment.
>
> We will clarify this distinction more explicitly in Section 4.2 of the final version. Overall, our design remains faithful to the core philosophy of DPO but introduces a lightweight and task-specific extension tailored for VAD.

---

> ### Author Response · Authors · 2025-08-06
>
> Dear Reviewer b99r,
>
> I hope this message finds you well. We greatly appreciate the insightful comments and suggestions you have provided regarding our paper. We have made every effort to thoroughly address each of your queries in our detailed response. As the discussion phase comes to an end, we remain open and eager for any further discussion that could help refine and improve our work. Your insights have been extremely helpful to us. Thank you once again for your time and effort.
>
> Best regards,\
> Authors of Submission933

---

> ### Comment · Reviewer_b99r · 2025-08-09
>
> Thank you for your responses. I am satisfied that they have addressed my concerns. I'm raising my rating to 6: Strong Accept. This is a great paper and is recommended for presentation at the venue.

---

> > ### Author Response · Authors · 2025-08-09
> >
> > Dear Reviewer b99r,
> >
> > We are truly grateful for your thoughtful engagement throughout the review process and for taking the time to revisit our work during the discussion phase. Your encouraging feedback and recognition mean a great deal to us.
> >
> > We deeply appreciate your constructive comments, which have helped us improve the clarity and rigor of our paper. It is very motivating to know that our revisions have addressed your concerns.
> >
> > Thank you again for your time, generosity, and kind support.
> >
> > Warmest regards,  \
> > Authors of Submission933

---

### Note · Authors · 2025-08-12

Dear AC and Reviewers,

We sincerely thank all reviewers and the AC for their time, constructive feedback, and engagement during the review and discussion phases.

**Summary of our work:**
We propose **VAD-DPO**, a direct preference optimization framework that diagnoses and mitigates co-occurrence–induced hallucinations in LVLM-based VAD. By constructing visually similar but semantically contrasting video pairs, VAD-DPO explicitly addresses object–label shortcut learning, encouraging models to ground predictions in scene semantics rather than statistical correlations.

**Strengths noted by reviewers:**
- Systematic analysis of co-occurrence-driven hallucinations in LVLM-based VAD, a critical yet underexplored failure mode.
- Comprehensive multi-level diagnosis with targeted probe sets, bridging analysis and mitigation.
- Effective, generalizable solution that improves anomaly detection and reasoning across diverse backbones.
- Clear, well-organized presentation with informative figures, tables, and ablations.

**Responses to reviewer feedback:**
- On *performance on probe sets*, we move the original analysis from the supplementary material into the main discussion and expand it to include domain-specific expert models.
- On *comparison with domain-specific models*, we add results for Holmes-VAU, Hawk, and training-free LVLM-based methods.
- On *generality on more LVLMs*, we demonstrate consistent gains on InternVL3-1B, exceeding even larger off-the-shelf InternVL3-8B.
- On *data scale*, we show that carefully constructed preference pairs preserve or enhance generalization on four unrelated video understanding benchmarks.
- On *evaluation details*, we fully describe our frame-level scoring pipeline, thresholds, and manual validation for both probe sets.
- On *related work and novelty*, we distinguish VAD-specific object–label co-occurrence hallucination from prior object hallucination work.

Following these clarifications, Reviewer **b99r** confirms concerns are addressed and raises the score to *Strong Accept*; **TxqF** and **6ZUx** confirm no remaining concerns; **BrW1** submits the acknowledgement and does not raise any new issues.

In the final version, we will ensure that all constructive feedback is thoroughly reflected to further improve the paper.

We thank the AC and reviewers once again for their thoughtful evaluations and constructive engagement.

Best regards, \
Authors of Submission 933

---

### Decision · Program_Chairs · 2025-09-17

**Decision:**

Accept (poster)

**Comment:**

This paper has received ratings of 4456 after the rebuttal. The paper proposes a framework called VAD-DPO to address hallucinations introduced by visual-textual co-occurrence in large vision-language models for video anomaly detection. The proposed approach analyzes this failure mode and improves the overall performance by focusing on semantic consistency instead of statistical shortcuts. The paper has conducted comprehensive evaluations across multiple datasets, and the approach has sufficient technical novelty to mitigate hallucinations in VAD. There were some weaknesses regarding the limited data scale and insufficient comparisons with domain-specific models. The paper has made significant contributions to understanding and addressing hallucinations in VAD. The authors have generally addressed reviewer concerns during the rebuttal. Thus, the AC recommends acceptance to NeurIPS.